# Sudden large-volume detachments of low-angle mountain glaciers – more frequent than thought?

Andreas Kääb[1], Mylène Jacquemart[2], Adrien Gilbert[3], Silvan Leinss[4], Luc Girod[1], Christian Huggel[5], Daniel Falaschi[6,7], Felipe Ugalde[8,9], Dmitry Petrakov[10], Sergey Chernomorets[10], Mikhail Dokukin[11], Frank Paul[5], Simon Gascoin[12], Etienne Berthier[13], Jeff S. Kargel[14]

[1] Department of Geosciences, University of Oslo, Norway
[2] Cooperative Institute for Research in Environmental Sciences, University of Colorado at Boulder, United States
[3] Université Grenoble Alpes, CNRS, IGE, Grenoble, France
[4] Institute of Environmental Engineering, ETH Zurich, Switzerland
[5] Department of Geography, University of Zurich, Switzerland
[6] Instituto Argentino de Nivología, Glaciología y Ciencias Ambientales, Mendoza, Argentina
[7] Departamento de Geografía, Facultad de Filosofía y Letras, Universidad Nacional de Cuyo, Mendoza, Argentina
[8] Geoestudios, San José de Maipo, Chile
[9] Departamento de Geología, Facultad de Ciencias Físicas y Matemáticas, Universidad de Chile, Santiago, Chile
[10] Faculty of Geography, M.V.Lomonosov Moscow State University, Moscow, Russia
[11] High-Mountain Geophysical Institute, Nalchik, Russia
[12] CESBIO, Université de Toulouse, CNES/CNRS/INRA/IRD/UPS, Toulouse, France
[13] LEGOS, CNES, CNRS, IRD, UPS, Université de Toulouse, Toulouse, France
[14] Planetary Science Institute, University of Arizona, Tucson, AZ, USA

*Correspondence to*: Andreas Kääb (kaeaeb@geo.uio.no)

## Abstract.

The detachment of large parts of low-angle mountain glaciers, resulting in massive ice-rock avalanches, have so far been believed to be a unique type of event, made known to the global scientific community first for the 2002 Kolka Glacier detachment, Caucasus Mountains, and then for the 2016 collapses of two glaciers in the Aru range, Tibet. Since 2016, several so-far unrecognized low-angle glacier detachments have been recognized and described, and new ones have occurred. In the current contribution, we compile, compare and discuss 20 actual or suspected large-volume detachments of low-angle mountain glaciers at ten different sites in the Caucasus, the Pamirs, Tibet, Altai, Alaska's St. Elias mountains, and the Southern Andes. Many of the detachments reached volumes in the order of 10–100 million $m^3$. The similarities and differences between the presented cases suggest that glacier detachments often involve a coincidental combination of factors related to lowering of basal friction, high or increasing driving stresses, concentration of shear stress, or low resistance to exceed stability thresholds

. Particularly, soft glacier beds seem to be a common condition among the observed events, as they offer smooth contact areas between the glacier and the underlying substrate, and are prone to till-strength weakening and eventually basal failure under high pore-water pressure. Surface slopes of the detached glaciers range between around 10° and 20°. This may be low enough to enable the development of thick and thus large-volume glaciers, while also being steep enough to allow critical driving stresses to build up. We construct a simple slab model to estimate ranges of glacier slope and width above which a glacier may be able to detach when extensively losing basal resistance. From this model we estimate that all the detachments presented occurred due to a basal shear stress reduction of more than 50%. Most of the ice-rock avalanches resulting from the detachments in this study have a particularly low angle of reach, down to around 0.1 (apparent friction angle), likely due to their high ice content and connected liquefaction potential, the availability of soft basal slurries and large amounts of basal water, as well as the smooth topographic setting typical for glacial valleys. Low-angle glacier detachments combine elements, and likely also physical processes of glacier surges and ice break-offs from steep glaciers. The surge-like temporal evolution ahead of several detachments and their geographic proximity to other surge-type glaciers suggests the glacier detachments investigated can be interpreted as endmembers of the continuum of surge-like glacier instabilities. Though rare, glacier detachments appear to be more frequent than commonly thought and disclose, despite local differences in conditions and precursory evolutions, the fundamental and critical potential of low-angle soft glacier beds to fail catastrophically.

# 1 Introduction

Eighteen years after the detachment of Kolka Glacier in the Russian Caucasus, the 17 July and 21 September 2016 detachments of two neighbouring glaciers in the Tibet's Aru range directed attention to a new type of glacier instability that had been rarely observed and little described before (Gilbert et al., 2018; Kääb et al., 2018; Tian et al. 2017). The detachment of Kolka Glacier on 20 September 2002 released $130 \cdot 10^6$ m$^3$ of ice and rock that claimed ~135 lives. Situated near a dormant volcano, Mt. Kazbek, it was long assumed that the Kolka Glacier catastrophe was unique and specific to the glacier's location (Haeberli et al., 2004; Huggel et al., 2005; Drobyshev, 2006; Evans et al., 2009b). The Aru twin glacier detachments – which released 68 and $83 \cdot 10^6$ m$^3$ of glacier ice without known conditions of high geothermal flux – have recently raised the questions of whether and where such events may have happened before or need to be expected in the future, what conditions allow low-angle glaciers to detach catastrophically from their beds, and what this means for mountain hazard management. The urgency of these questions is highlighted by the fact that several detachments similar to the Aru events, though smaller, have been detected subsequently (Falaschi et. al., 2019; Paul 2019; Jacquemart et al., 2020).

In contrast to glacier detachments, glacier surges are an extensively studied, though still not fully understood type of glacier instability. Characterized by unusually high ice-flow speeds of up to tens of metres per day over large parts of a glacier, glacier surges last weeks to several years (Harrison and Post, 2003; Jiskoot, 2011; Harrison et al., 2015; Truffer et al., 2021). Clusters of surge-type glaciers are found in many mountain regions around the world (Sevestre and Benn, 2015). The lowering of basal

glacier friction that is associated with surging involves abnormally high water pressure, change in the thermal regime, and/or responses of subglacial till to increasing shear stress and water input (Clarke et al., 1984; Kamb, 1987; Truffer et al., 2000; Fowler et al., 2001; Murray et al., 2003; Frappe and Clarke, 2007; Sevestre et al., 2015; Benn et al., 2019).

A second, well-known type of glacier instability happens over a wide range of magnitudes, from icefalls at steep glacier fronts to large ice avalanches, when partial or entire ice volumes suddenly break off from hanging glaciers that are typically steeper than around 30° (Alean, 1985; Huggel, 2009; Faillettaz et al., 2015). The latter empirical value from literature offers a slope threshold to separate the definitions of ice avalanches from glacier detachments. Impacts associated with ice avalanches, which typically have volumes much smaller than $1 \cdot 10^6$ m³, are usually limited to a few kilometres, unless the failed ice transforms into a highly mobile mass flow through liquefaction and incorporation of wet sediments or liquid water in the path (Petrakov et al., 2008; Evans and Delaney, 2015). Failure conditions and triggering factors of such steep ice avalanches typically include glacier geometry (steep ramp-type glaciers, bedrock edges), bedrock topography (e.g. convex bed), atmospheric events (e.g. temperature increase), increasing accumulation rates, ice-thermal conditions (e.g. frozen base or changes therein), instabilities of the underlying bedrock that take with them ice resting on it, or seismic events (Alean, 1985; van der Woerd et al., 2004; Huggel, 2009; Fischer et al., 2013; Faillettaz et al., 2015).

Compared to the above two types of glacier instability, sudden large-scale detachments of mountain glaciers, primarily occurring at low bed slopes, are much less frequent. However, due to a mobility at least as high as that of ice avalanches, combined with large volumes, glacier detachments can constitute a severe threat to communities and settlements located in remote areas.

In general, mass movements that result from sudden slope failures in ice- or snow-rich mountain environments are particularly mobile, leading to strongly increased run-out distances compared to ice/snow-free conditions (Petrakov et al., 2008; Huggel, 2009; Schneider et al., 2011; Evans and Delaney, 2015). Frictional heating melts ice and snow components, which are either part of the initial slope failure or incorporated along the avalanche path. Liquid water, embedded in the glacier and sediments before failure, can amplify the avalanche mobility. Also ice and snow surfaces, in case the avalanche travels over those, are able to reduce basal friction. Here, we define sudden large-volume glacier detachments through their initiation, while the eventually resulting ice-rock avalanches might be similar to those resulting from other high-mountain slope instabilities. By terming these events *low-angle glacier detachments*, we follow the suggestion by Evans and Delaney (2015) who describe the Kolka case as large-scale detachment of a valley glacier. Other authors, for instance, called these failure events glacier slides, in reference to landslides (Petrakov et al., 2008), or glacier collapses (Kääb et al., 2018).

The following selection focusses on detachments $\gg 1 \cdot 10^6$ m³ and from glaciers with surface slopes of less than around 20°, i.e. focussing on glaciers from which large-volume detachments are not expected. We are well aware that it can be reasonable to include events beyond the limits of these criteria in analyses, depending on the goal of the investigation, and that similar events beyond the limits of these criteria could involve the same mechanisms that are discussed here. The main scientific purpose of

this paper is to provide an overview of all known glacier detachments, either by summarizing existing detailed studies, or by providing such details for the first time (see Suppl. Tab. S1 for existing studies and new contributions by the present work). We aim to show and discuss the relation of low-angle glacier detachments to glacier surges and the continuum of high-mountain ice and rock instabilities, such as normal avalanches. The main applied purpose of our study is to make experts involved in high-mountain hazard management aware of the so far little recognized possibility for glacier detachments, and to discuss related potential key indicators and how climate change could factor into the mechanisms.

## 2 Ice-rock avalanches and glacier surges

In this section we draw some comparisons of low-angle glacier detachments to (1) more typical types of glacier ice and ice-rock avalanches and (2) glacier surges. The detachment process sequence combines elements of both of these but in a combination and under conditions that are distinct.

Glacier detachments lead to ice-rock avalanches, but ice-rock avalanches usually are be the result of different other initial types of slope failures and event cascades. A wide range of magnitudes, avalanche compositions and impacts have been observed (Schneider et al., 2011). In this section, we exemplify the diverse characteristics of ice-rock avalanches and the resultant mass flows in order to contrast them to sudden large-volume detachments of mountain glaciers. We use the extensive data collection from Schneider et al. (2011) as background data set for our study, extended by events from Petrakov et al. (2008). In Fig. 1, each event of this combined data set is plotted as a gray circle according to its horizontal reach (L), elevation difference (H) and detachment volume (V). The size of the circles in Fig. 1a indicates the event volumes. The ratio H/L is the apparent friction coefficient or the angle of reach, also called "Fahrböschung", calculated from the uppermost scarp of the slope failure to the lowermost part of the mass movement deposits. Dark grey circles mark the following examples, standing out mainly by type, volume and angle of reach:

- In the 1970 Huascaran (Peru) event (and a similar event in 1962), a rock-wall failure triggered by a M7.9 earthquake, incorporated large amounts of ice from above and below it, leading to a highly mobile and far reaching rock-ice avalanche of $80 \cdot 10^6$ m$^3$ that claimed up to 20'000 lives (e.g., Evans et al., 2009a). In Fig. 1, we use the solid deposits of the avalanche to define its reach and elevation difference, neglecting that a subsequent water/mud flood travelled much farther (indicated by * symbols in Fig. 1). The 1970 Huascaran avalanche is one of the largest, farthest reaching, and the by far deadliest known ice-rock avalanche.

- A rock failure in the Chilean Andes in 1987 incorporated ice, snow and water which transformed the avalanche into a debris flow of $15 \cdot 10^6$ m$^3$ that sped down the Estero Parraguirre valley and killed more than 37 people (Hauser, 2002). In our collection, this is the farthest-reaching event (lowest angle of reach; H/L ~0.12) that did not originate as a glacier detachment.

-   The 1964/65 Allen Glacier event, Alaska, is an example for a very large rock avalanche, likely triggered by an earthquake, that was able to run out for an unusually long distance because it landed on a glacier (Post, 1968). Neighbouring glaciers
show similar rock deposits from the same time, and also later earthquakes caused comparable rock avalanches that travelled far over low-angle glaciers (e.g., Shugar et al., 2012).

-   In 1895, $5 \cdot 10^6$ m$^3$ of at least partially cold-based ice sheared off 40°-steep bedrock from Altels Glacier in the Swiss Alps (glacier surface slopes indicated as numbers within the circles in Fig. 1a). The resulting ice avalanche rushed up the opposite
side of the valley and thus did not reach its maximum runout distance (indicated by → symbols in Fig. 1; Faillettaz et al., 2011). The Altels event is an example of a very large, pure ice avalanche, stemming from the detachment of a very steep
glacier. Until it sheared off, the glacier was probably held in place by transverse bedrock riegels and cold patches/zones where it was frozen to its bed (Wagner, 1996).

-   A number of ice-rock avalanches have occurred from different locations on Iliamna volcano, Alaska, the last of which is documented for June 2019 (Toney et al., 2020). Failure surfaces were typically on the order of 40°, and volumes reached
up to around $20 \cdot 10^6$ m$^3$ (Caplan-Auerbach and Huggel, 2007; Huggel et al., 2007). These events show that enhanced geothermal heat fluxes can be involved in causing ice-rock avalanches.

Above, we exemplify ice-rock avalanches other than glacier detachments in order to put the avalanches resulting from glacier detachments into the context of high-mountain mass movements. In the following, we also put glacier detachments briefly in
the context of glacier surges. An obvious difference between surges and glacier detachments is that the bed of surging glaciers does not fail catastrophically. A substantial body of research is available about glacier surging, covering among others surge
cycles and phases, and thermally and/or hydrologically driven surge mechanisms (see references in the Introduction). Some observations of surging mountain glaciers or their surge-like movements fall outside the norm of the majority of surges, but
will become of interest for some glacier detachments contained in this contribution: While the regional pattern of known glacier surges exhibits geographical clusters (Sevestre and Benn, 2015), surge-like events are sometimes found far outside the
known surge clusters, as was for example the case for the speed-up of Belvedere Glacier, Italian Alps, in the early 2000s (Haeberli et al., 2002; Kääb et al., 2004; Harrison et al., 2015; Truffer et al., 2021) (cf. Tsambagarav detachment, Section 3.4).
Ice flow speeds associated with surges are typically one to two orders of magnitude higher than pre-surge speeds, up to several tens of metres per day, or about 5-6 orders of magnitudes slower than ice-rock avalanches. But there are known surges that
reached speeds of up to 1 km/h, i.e. only about 2 orders of magnitude slower than the avalanches (Section 3.6.3; Zhang, 1992). Furthermore, disintegration of surging (and non-surging) glaciers, which did not lead to ice-rock avalanches, have also been
reported (e.g., Milana, 2007; Wang et al., 2020). Finally, another non-frequent behaviour of surge-type glaciers is extreme surface bulging that has been observed as a consequence of polythermal ice structure (Clarke and Blake, 1991) (cf. e.g. Flat
Creek detachments, Section 3.7).

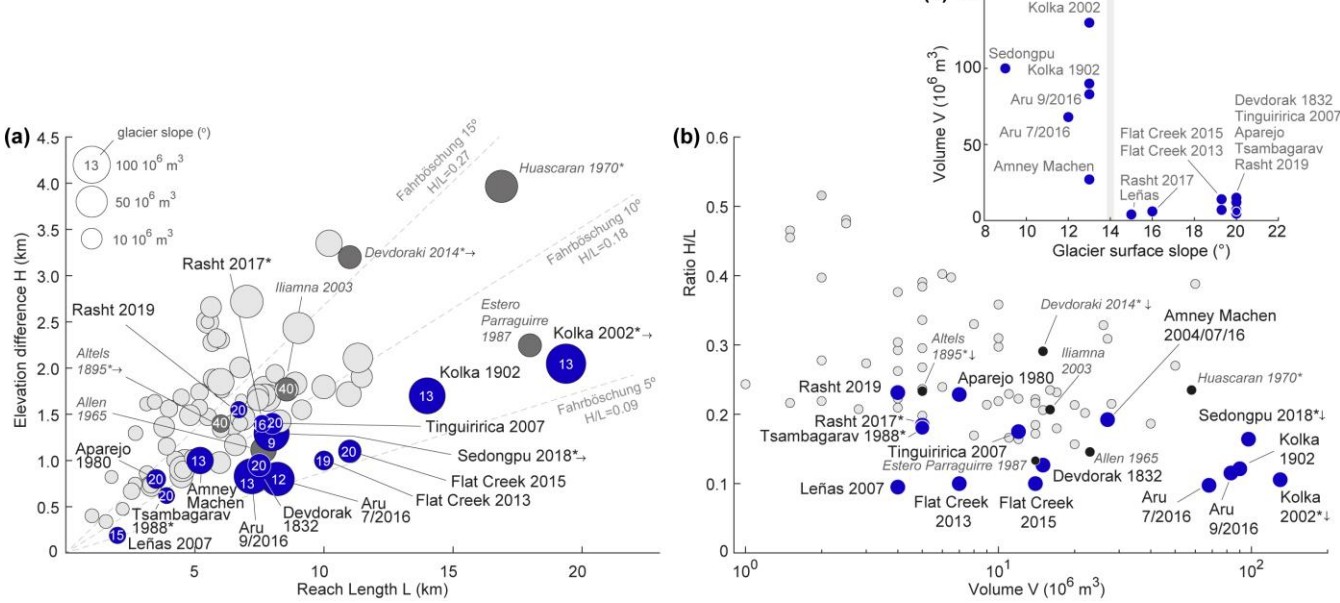

**Figure 1: (a) Known avalanches of rock-ice mixtures plotted by elevation drop H versus reach L. The event volumes are indicated by circle size (see legend to the upper left of panel a). Blue circles are sudden large-volume detachments of low-angle mountain glaciers, with the surface slope of the detached glacier parts given inside the circles. The grey circles are other ice-rock avalanche events, with dark grey events mentioned specifically in the text to illustrate different event types collected in this figure. The diagonal dashed lines indicate the angle of reach, Fahrböschung, at 5°, 10° and 15°. (b) Events from (a) plotted by reach angle (H /L) versus avalanche volume V. Note the logarithmic scale of the x-axis (volume). Most events stem from Schneider et al. (2011), extended by data from Petrakov et al. (2008) and the present study. * after the event name indicates that a mud or debris flow continued from the rock-ice avalanche deposits, which is not considered in the calculation of the reach. Arrows behind the event name indicate that the avalanche was stopped by some obstacle and would otherwise have travelled farther. The direction of the arrow indicates the direction in which the event symbol would have shifted in the plot without the obstacle. (c) Glacier detachment volume against glacier surface slope. The grey vertical bar indicates a very rough boundary between the glacier slope of very large and smaller detachment volumes.**

# 3 Glacier detachment events

In the following, we summarize events that we categorize as glacier detachments, i.e. large-volume ice-rock avalanches from the sudden failure of low-angle parts of mountain glaciers (blue circles in Fig.1). Based on previously published findings from several of the events, we focus our descriptions specifically on disposition factors that could contribute to or hint at the presence of particularly low basal friction and high driving stresses. These factors include soft sediments, polythermal ice conditions, abnormal geothermal heat flux, high water input and basal water pressure, glacier surging, additional ice or rock loading, or significant steepening in surface slope. As it has been shown that thermal conditions can also play a role in the detachments of glaciers (Gilbert et al., 2018; Jacquemart et al., 2020), we also try to evaluate permafrost conditions for each event.

We sort the following events by regions, and proceed within the regions from short summaries of well-documented events to more detailed descriptions of not or little studied cases. Discussions of individual events, e.g. regarding pre-satellite era events or a possible recurrence by glacier recovery, are included in the respective subsections, whereas the main discussion Section 4 focuses on the overall comparison between events.

## 3.1 Mount Kazbek, Caucasus mountains

### 3.1.1 Devdorak, 18[th] century, 19[th] century, 2014

A number of suspected glacier detachments and rock-ice avalanches happened around Mt. Kazbek at the border between Russia and Georgia. During the 18-19[th] centuries, surges of the Devdorak *(*Georgian name Devdoraki also used in literature) Glacier on the north-eastern flank of Mount Kazbek (Fig. 2; Tab.1) were moving down the Amilishka River (Kabakhi River in its lower part), which drains the glacier. Surge-like advances were recorded in 1776, 1778, 1785, 1808, 1817, and 1832 (Zaporozhchenko and Chernomorets, 2004). On August 13, 1832, parts of the surging Devdorak Glacier tongue detached and the subsequent ice-rock avalanche blocked the main Terek valley, an important transportation route between Russia and Georgia (Petrakov et al., 2008). Eyewitness of the 1832 event, engineer-captain Grauert estimated a volume of $15.5 \cdot 10^6$ m$^3$ of ice and rock mass blocked the Dariali Gorge of the Terek valley (Zaporozhchenko and Chernomorets, 2004). The causes and mechanisms of this (and other) detachments are not well known, and vary throughout the literature. Ice-rock avalanching onto the glacier, overloading and associated increase in subglacial water pressure could well have played a role.

Except for their source area, these at their time well known "Kazbek blockages" followed the same avalanche path as the 2014 Devdorak event described in the following. In 2014, parts of a rock wall and overlying hanging glaciers failed from Mount Kazbek (Figs. 2-3). The resulting highly mobile rock-ice avalanche of $2\text{-}5 \cdot 10^6$ m$^3$ rushed down the Amilishka / Kabakhi Valley and blocked the main road between Russia and Georgia, killing 9 people (Chernomorets et al., 2016; Tielidize et al., 2019). This event might also have had a longer runout (Fig. 1) but was stopped by a sharp 90°-turn at the confluence of the Devdoraki gorge and the Dariali (or Terek) gorge. The 1832 Kazbek blockage is described as a glacier detachment in the sense of the present contribution in older literature, but more recent interpretations, not least based on the 2014 event, suggest that the 1832 and other such events at the site might have started as rock failures rather than surges (Chernomorets, 2014, Chernomorets et al., 2016). We list the Kazbek blockages as potential glacier detachments here, but stress that it remains uncertain what kind of event they actually were.

Currently, the entire narrow lower part of Devdorak Glacier has a surface slope of around 23°, its tongue closer to 17°. The volcanic nature of Mount Kazbek, the documentation of a number of violent mass flows from the mountain in the past (Chernomorets et al., 2007), field visits, and visual analysis of very high resolution satellite images and terrestrial photos all suggest highly erodible rock and an abundance of fine sediments at many places on Mount Kazbek. On all high-resolution satellite images available since around 2002 (GoogleEarth, Bing Maps, Maxar, Pleiades), Devdorak Glacier appears heavily

crevassed, and is partly covered by fine sediments, likely deposited by mass movements from the surrounding mountain flanks. Since 2015, a glacier directly south of Devdorak Glacier has been overriding the main tongue of Devdorak glacier in a surge-

like destabilisation, perhaps triggered by the 2014 Kazbek/Devdorak rock-ice avalanche that overran it (Fig. 3). Possibly as a consequence of the tributary surge, Devdorak Glacier itself is currently also advancing (Dokukin et al., 2020). A 1-km global

permafrost model (Obu et al., 2019), not particularly tuned for mountain permafrost, though, suggests the elevation of the Devdorak Glacier tongue is roughly at or below today's lower boundary of the discontinuous permafrost zone in the region.

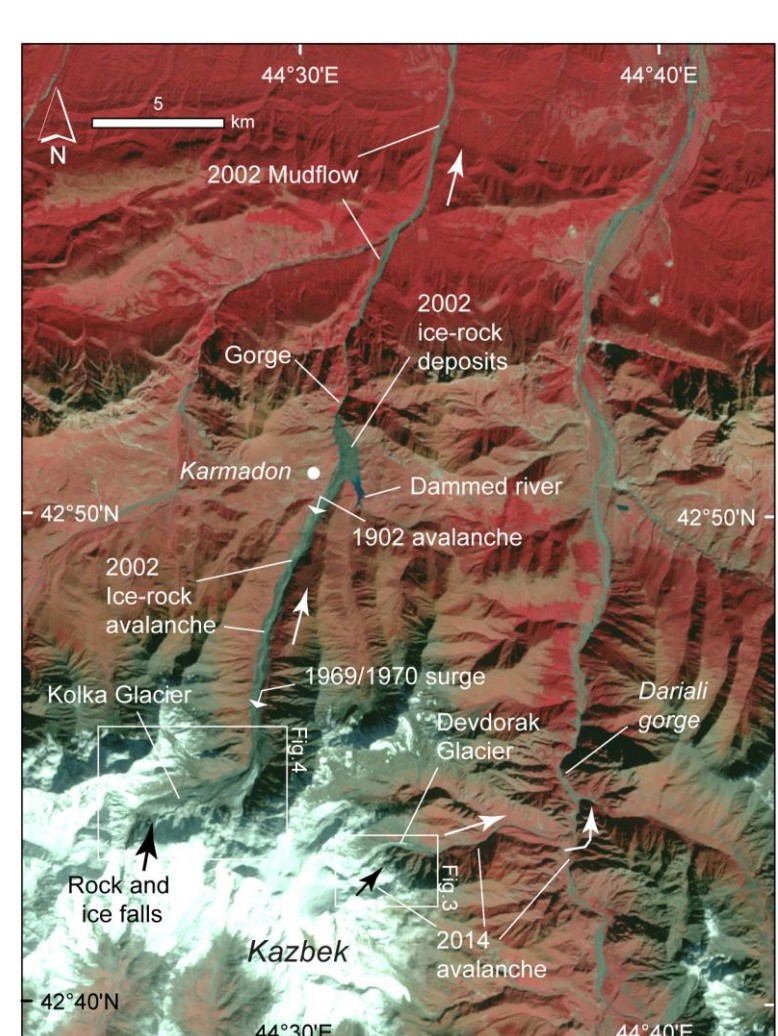

**Figure 2: Overview over Mount Kazbek, Russia/Georgia, Caucasus Mountains, with elements of the 2002 Kolka/Karmadon and 2014 Devdoraki rock-ice avalanches indicated. Satellite image: Landsat, 6 Oct 2002 (credit:**
**USGS).**

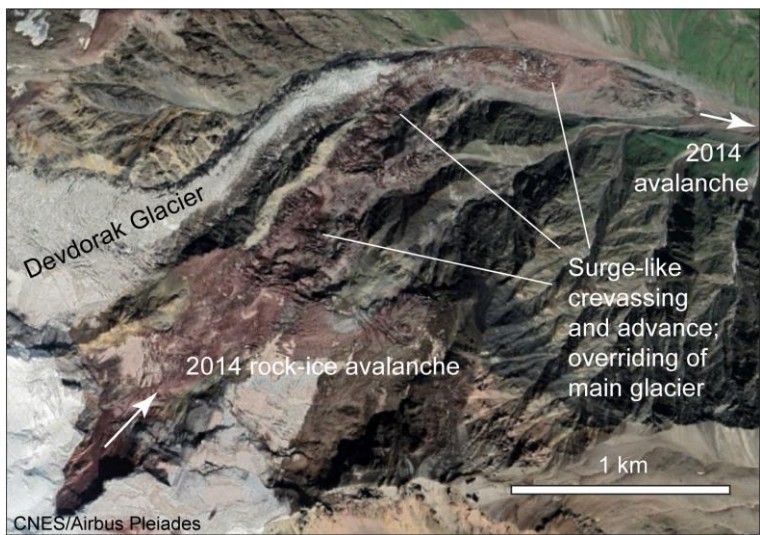

Figure 3: Lower part of Devdorak Glacier and northeastern flank of Mount Kazbek. The position of the image section is indicated in Fig. 2. Satellite image: Pleiades, © Airbus, 19 Aug 2019.

### 3.1.2 Kolka, 1902 and 2002

The $130 \cdot 10^6$ m³ Kolka Glacier detachment (Figs. 2 and 4, Tab. 1) of 20 September 2002 has been described and discussed in several studies (Kääb et al., 2003; Haeberli et al., 2004; Kotlyakov et al., 2004; Huggel et al., 2005; Drobyshev, 2006; Evans et al., 2009b). During these investigations, it became clear that a similar event must have already happened at least once, at the beginning of July 1902 (and probably also around 1700), whereby the glacier tongue detached after a roughly two-week long surge-like advance. Heavy rain and snow melt may have played a role in triggering the 1902 advance. Damming or erosion by water could then have caused the actual detachment and subsequent rock/mud/ice flow (Drobyshev, 2006; Petrakov et al., 2008; Kotlyakov et al., 2010b). The descriptions given in these references and their sources clearly describe an event that qualifies as a detachment in the sense of the present contribution. The glacier started surging again in autumn 1968 and advanced by 4 km with speeds of up to 220 m/day (Fig. 4), but without catastrophic consequences (Hoinkes, 1972; Rototayev et al., 1983). We do not know which differences in conditions, compared to 1902 and 2002, caused the glacier to not detach in 1970. Several other glaciers in the Caucasus are known to have surged in the past (Kotlyakov et al., 2010b). We found that Kolka Glacier had a low surface slope of around 13° prior to the 2002 detachment. Evans et al (2009b) estimated a bed slope of 9° after the detachment. Over the course of several weeks before the 2002 detachment, perhaps triggered by earthquakes (Kotlyakov et al., 2004), heavy rock and ice falls from the northern flank of the Kazbek massif deposited several $10^6$ m³ of material on the glacier. During this period of mass-wasting activity, the glacier changed in unusual ways: it bulged and became heavily crevassed (Fig. 4), it developed a scarp at the location of the later detachment, and supraglacial ponds formed

(Kotlyakov et al., 2004; Evans et al., 2009b; Kotlyakov et al., 2010b). Unusually high geothermal heat fluxes underneath the glacier (fumaroles and sulphur smell were reported from the glacier bed shortly after detachment), the impact energy of a large rock/ice fall, and successive loss of shear stress due to excess water pressure have been proposed as possible factors and ultimate triggers of the 2002 detachment (Kotlyakov et al., 2004; Evans et al., 2009b; Kotlyakov et al., 2010b). Similarly, the additional loading of Kolka Glacier from the rock and ice falls has more recently been proposed to have increased the basal shear stress until it exceeded a frictional threshold given by the glacier bed material, topography, and hydraulic conditions (Kääb et al., 2018). After the detachment, lakes were visible on the Kolka Glacier bed, pointing to the involvement of large amounts of subglacial water in the detachment. The high mean avalanche velocities of 50-80 m/s (Huggel et al., 2005) suggest availability of large amounts of water and/or saturated fine-grained till.

The Kolka Glacier tongue is, like Devdorak Glacier, roughly at the lower elevation of the regional discontinuous permafrost zone, and the glacier must have been temperate throughout, except for the likely polythermal, or even cold steep hanging glaciers avalanching from the northern flank of the Kazbek massif onto its surface. The ice-rock avalanche resulting from the 2002 detachment is described in detail in the above literature about the event, but we want to draw attention to the streamlined debris stripes, in the following called debris stripes, that were visible in the detachment zone after the event (Fig. 4d; Petrakov et al., 2004; Huggel et al., 2005), because similar patterns have also been found in several of the other cases of this study. Currently, the detached glacier depression is refilling and the glacier is gaining mass again. The fast recovery of the glacier volume lost by the 1970 surge, and again after the 2002 detachment, is associated with Kolka Glacier's positive mass balance, which stands in stark contrast to the predominant strong glacier shrinkage in the Caucasus Mountains (Kutuzov et al., 2019; Zemp et al., 2019). Kolka Glacier had already reached almost 50% of its pre-detachment volume by 2017 and is projected to accumulate 60–70% of its pre-detachment volume by 2025 (Petrakov et al., 2018; Aristov et al., 2019).

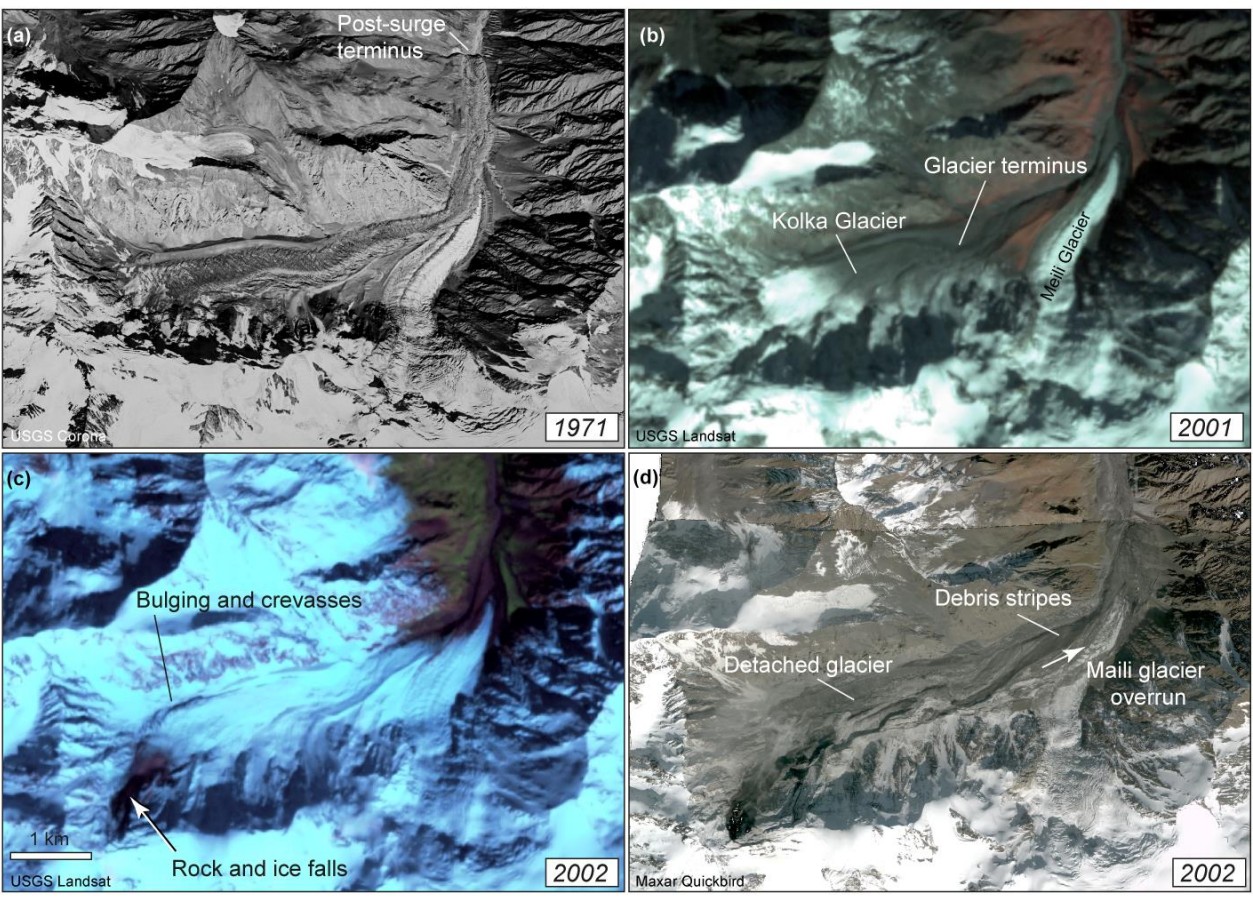

**Figure 4: Kolka Glacier, Mount Kazbek. (a) Kolka Glacier surged in 1969/70. Satellite image: Corona, 20 Sep 1971**
**(credit: USGS). (b) Landsat image, 3 Oct 2001 (credit: USGS). (c) Days and weeks before the 2002 detachment, rock and ice falls/avalanches were observed onto the glacier, and the left lateral margin of the glacier was bulged and heavily**
**crevassed. Satellite image: Landsat, 20 Sep 2002, a few hours before detachment (credit: USGS). (d) Quickbird satellite image (© Maxar), 25 Sep 2002, 5 days after detachment. The position of the image sections is indicated in Fig. 2. The**
**location of debris stripes in avalanche direction is indicated.**

## 3.2 Rasht, Pamir/Tajikistan

In 2017 and 2019, two glacier detachment events happened on the north side of the Peter the First Range (or Petra Pervogo
Range, or Peter the Great Range), in the  Pamir Mountains of Tajikistan. The resulting masses of both events travelled north
into the Rasht Valley through which the Surkhob River flows that later forms the Vakhsh River (Fig. 5).

### 3.2.1 Event of 2017

The 2017 event, first mentioned in Dokukin et al. (2019), happened between 10 and 11 July 2017 (Planet images). A glacier of roughly 1000 m length and 240 m width detached (upper scarp ca. at 3600 m a.s.l) and the resulting ice-rock avalanche flowed down a narrow valley towards the village Tojikabod in the Rasht Valley (Figs. 5 and 6, Tab. 1). In satellite imagery taken shortly after the event (Planet, Maxar), ice remains can be recognized over a horizontal distance of about 8 km, down to an elevation of about 2160 m a.s.l. Beyond this point, a considerable debris/mud flow must have continued for another 2 km or so. The detached glacier had a surface slope of around 16° (High Mountain Asia DEM; Shean, 2017). We measured increased surface speeds of up to 5.8 m/day in early July 2017 (compared to a few dm/day in 2016; velocities from repeat Planet data) and detected unusual lateral crevasses delineating the later detachment area in images from as early as May 2017 (Fig. 6). The glacier is likely not surrounded by permafrost (Obu et al., 2019).

To roughly estimate the event volume we derive the ice thickness along the center flow line of the glacier based on an estimated basal shear assuming a driving stress of $1.2 \cdot 10^5$ Pa as suggested for mountain glaciers, a slope of 16°, and a form factor of 0.8, which then results in a thickness of around 60 m (Cuffey and Patterson, 2010). Multiplying half of this depth (i.e. assuming a triangular cross-section) with the detached area (ca. 200,000 m$^2$) gives a first volume estimate of roughly 6 $10^6$ m$^3$. Whereas satellite images after detachment suggest that much of the glacier bed might actually have a triangular cross-section, it may have been more shallow in the lowermost and uppermost parts. As an order of magnitude, we suggest a detachment volume of $5 \cdot 10^6$ m$^3$ and assign a conservative error of $\pm 1 \cdot 10^6$ m$^3$ to this estimate.

The cirque from which the 2017 event originated was also the source of other slope instabilities over recent years. Another (much smaller) ice-rock avalanche from a neighbouring glacier occurred between 15 and 24 July 2016 (dates from Planet images), with a horizontal reach of roughly 5.5 km. A large debris flow descended the same valley in late August 2016, starting from the same cirque, likely entraining deposits of the July 2016 avalanche, and reaching the Surkhob River 19.4 km downstream where it destroyed several buildings, bridges and agricultural fields.

### 3.2.2 Event of 2019

Between 2 and 3 August 2019 a second glacier, ca. 14 km to the west of the one that collapsed in 2017, detached (Figs. 5, 7). This glacier was slightly smaller than the 2017 one, and had a surface slope of around 20°. The resulting ice-rock avalanche travelled north down a narrow valley towards the Rasht Valley, over a horizontal reach of 6.5 km and a vertical drop from 3350 (upper scarp) to 1850 m a.s.l. Superelevations of the ice-rock avalanche path of up to almost 200 m above valley bottom suggest high avalanche speeds (McClung, 2001). We estimated the detached volume in the same way as described for the 2017 event (section 3.2.1) and computed a volume of $4.5 \pm 1 \cdot 10^6$ m$^3$ (glacier area ca. 230,000 m$^2$; centreline depth 50 m). Based on pre- and post-event WorldView stereo DEMs Leinss et al. (2020) estimate a maximum erosion depth of 90 m and a detachment volume of $8\text{-}9 \cdot 10^6$ m$^3$, i.e. significantly more than our rough general model.

This glacier also showed increased sliding speeds and crevassing around the later detachment area for at least 2-3 weeks before the failure. For the end of July 2019 we found surface speeds of roughly 2.5 m/day, a marked increase compared to roughly < 0.1 m/day during 2017–2018 (repeat Planet data). The glacier did not show any visual signs of destabilisation between 2015 and 2018 (Planet images). In 2007 (Maxar; see Supplement) the glacier looked heavily crevassed, possibly an indication of a surge-like advance. This condition is still visible in Landsat data 5-6 years later, though less certain due to the lower resolution of Landsat data (no other data is available to us between 2007 and 2015). Landsat data also suggest that the glacier experienced a similar advance in the early 1990s. Under the limitation of the reduced spatial resolution of the Landsat data, however, we do not find signs of a large detachment event or large ice-rock avalanche. Nevertheless, we draw attention to a surprisingly vegetation-free landform visible downstream of the 2019 detachment in pre-event imagery (Fig. 8). The lack of vegetation, the streamlined microtopography, and zones of rough and chaotic microtopography that resemble avalanche or debris flow deposits (Fig. 8; Supplement), led us already before the 2019 event to interpret this landform as a possible geomorphological imprint of an earlier ice-rock avalanche (Kääb, 2019). Meanwhile, this landform has been overrun by the August 2019 ice-rock avalanche, leaving similar new forms, and suggesting that the landform now buried could have originated from a similar detachment event, probably before 1961 (year of earliest Corona satellite image). Between the 1961 Corona images and very high-resolution images from just before the 2019 avalanche no significant changes are visible on the landform.

Very-high resolution satellite images over the Peter the First Range suggest abundance of weak bedrock and fine sediments. All over the range, signs of large debris flows, rock avalanches or ice-rock avalanches are visible in high resolution satellite images (Maxar, CNES/Airbus, Planet; Leinss et al, 2020). The Pamirs are known to be very geomorphologically active, with a number of associated hazards (Mergili et al., 2012; Gruber and Mergili, 2013; Strom and Abdrakhmatov, 2018) and a cluster of surge-type glaciers (Kotlyakov et al., 2008; Kotlyakov et al., 2010a; Gardelle et al., 2013; Sevestre and Benn, 2015; Lv et al., 2019; Goerlich et al., 2020). A number of glaciers in the Peter the Great Range were surging at the time of writing or have done so in the recent past (Fig. 5). Didal Glacier (ca. 12° steep) surged around 1995, and again during the winter 2015/16 where it advanced by 2.5 km over a few months. In the valley below Didal Glacier we note lack of vegetation in the valley and landforms that could well stem from a former ice-rock avalanche. Kotlyakov et al. (2010a) mention a 2.2 km long ice avalanche from Didal Glacier in 1974. In comparison, the bottom of the valley through which the 2017 ice-rock avalanche descended (section 3.2.1) was partly tree covered before, suggesting that no event like the 2017 one has happened there in the recent past.

Even if not documented in detail in an internationally accessible format so far, to our best knowledge, both the 2017 and 2019 detachments and their downstream effects were very likely noted by the local communities as the lowermost ice/rock deposits stopped not far from settlements, agricultural fields, and pastures, and very-high resolution images (Maxar) show that flooding happened close to houses and two irrigation channel bridges were partially destroyed.

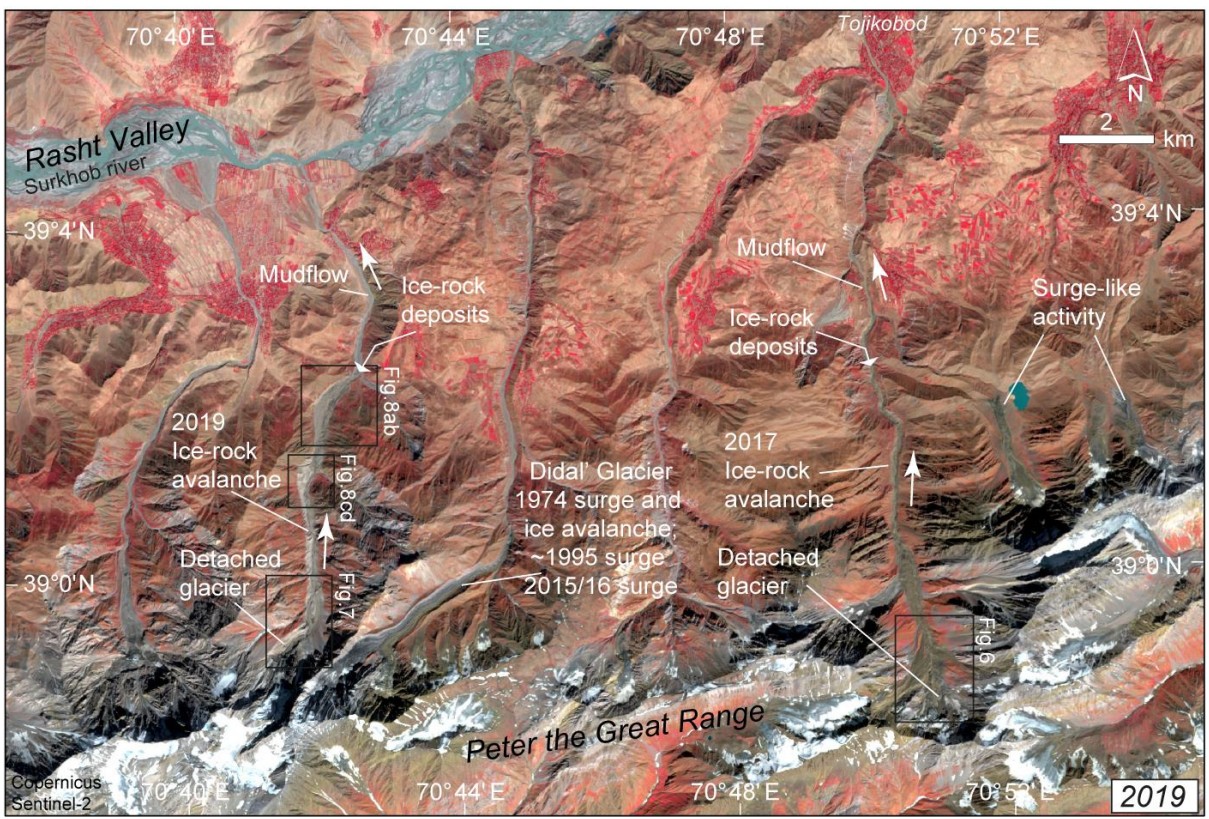

**Figure 5: Rasht Valley and Peter the Great Range, Tajikistan. Locations of the 2017 and 2019 ice-rock avalanches are indicated. Satellite image: Sentinel-2, 19 Sep 2019 (credit: Copernicus Sentinel data).**

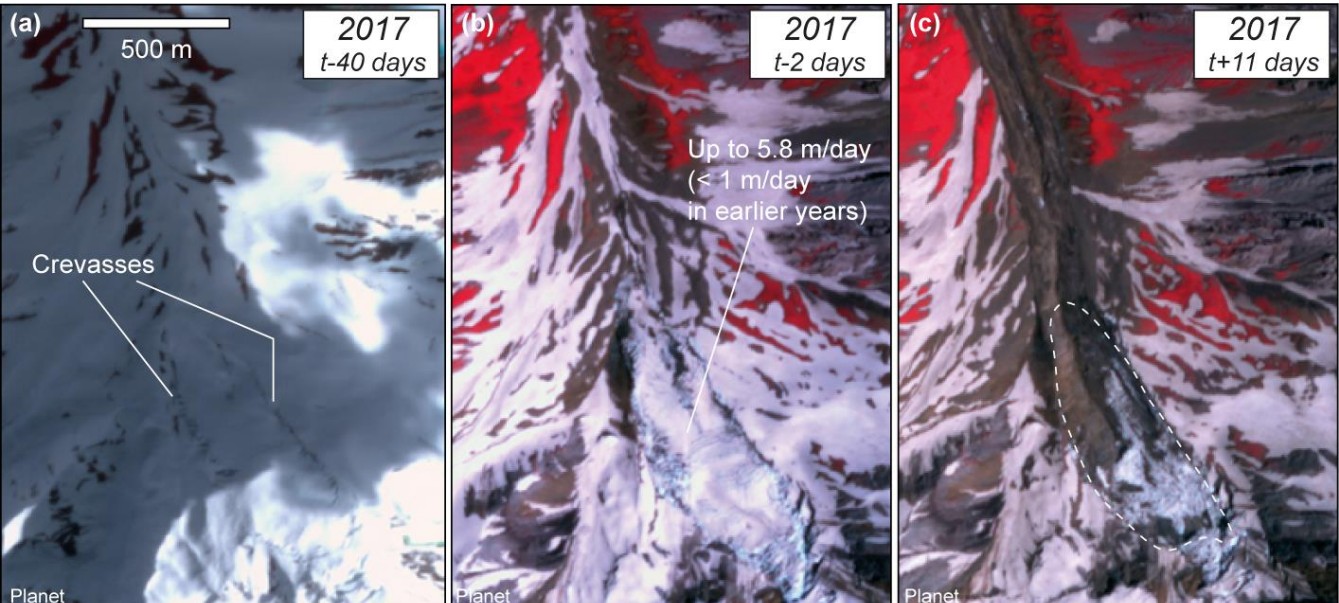

**Figure 6: Planet images over the Rasht Valley glacier detachment of around 10 July 2017 (=t). (a) 31 May 2017, (b) 8 July 2017, (c) 21 July 2017. Abnormal marginal crevassing and enhanced speeds were visible several weeks before the detachment. See Fig. 5 for location. (Satellite images © Planet).**

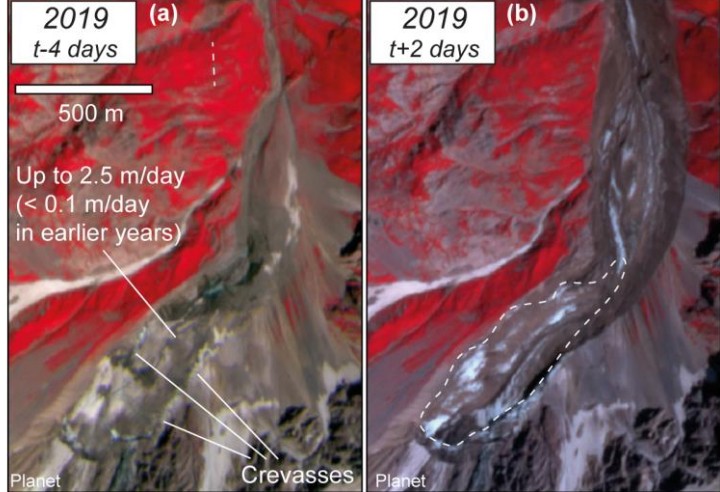

**Figure 7: Planet images over the Rasht Valley glacier detachment of around 2 Aug 2019 (=t). (a) 29 Jul 2019, (b) 4 Aug 2019. See Fig. 5 for location. (Satellite images © Planet).**

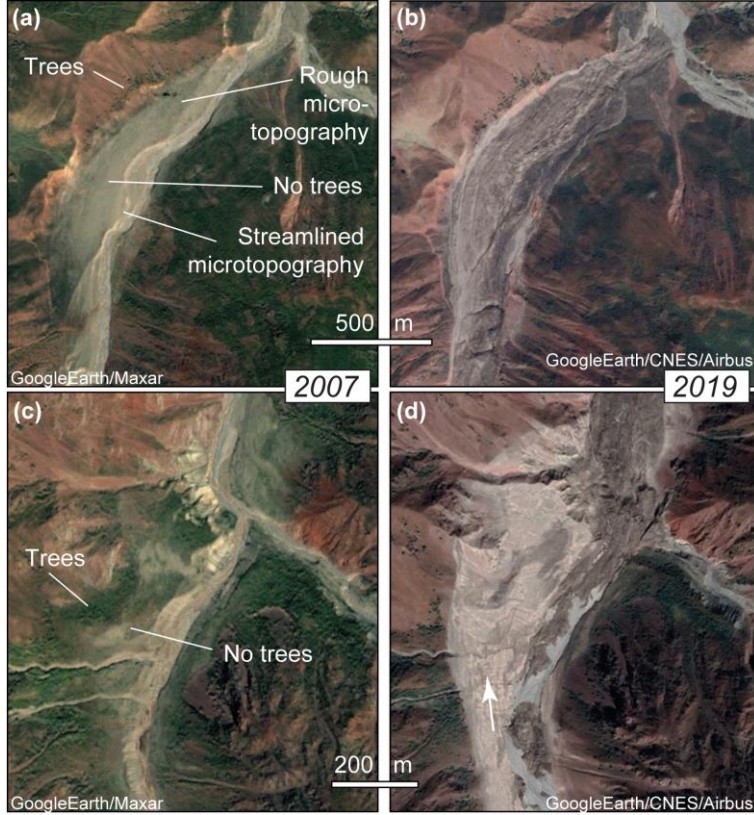

Figure 8: Location of the ~2 Aug 2019 Rasht Valley ice-rock avalanche. See Fig. 5 for image location. (a) and (c): 30 Jul 2007 (© Google Earth and Maxar). (b) and (d): 20 Aug 2019 (© Google Earth and CNES/Airbus). Before the 2019 event, traces of a potential former large mass flow were visible (lack of trees in the valley, sparse vegetation, debris stripes).

## 3.3 Aru, 2016, western Tibet

On 17 July 2016, a massive volume of glacier ice detached from the lower part of an unnamed glacier in the Aru Range (Rutok County, China) in the western Tibetan Plateau (termed Aru-1). The fragmented ice mass ran out 6 km beyond the glacier terminus, killing nine herders and hundreds of their animals, and reached the Aru Co lake (Tian et al., 2017; Kääb et al., 2018). The ice debris covered 8–9 km$^2$ and a volume of $68 \cdot 10^6$ m$^3$ was calculated for the detached glacier part. On 21 September 2016, a second glacier (Aru-2) detached just a few km south of Aru-1 (Fig. 9). Similar to the July event, the glacier ice fragmented and transformed into a mass flow. The glacier debris in the second detachment covered 6–7 km$^2$ with a detached glacier volume of $83 \cdot 10^6$ m$^3$. The maximum glacier thicknesses that detached were around 115 m (Aru-1) and 145 m (Aru-2),

and maximum deposit thicknesses were around 25 m (Aru-1) and 80 m (Aru-2) (Kääb et al.,2018). The mean speeds of the Aru ice-rock avalanches were estimated to 30–50 m s$^{-1}$, with maximum speeds of 70–90 m s$^{-1}$ (Kääb et al., 2018). (For a satellite image of the current situation of the Aru glaciers and the deposits, see the Supplement).

Glaciers in the wider region around the Aru range are part of the Karakoram – West Kulun Shan – Eastern Pamir anomaly (Treichler et al., 2019) and experienced a slight increase in thickness of around 0.20– 0.30 m/a water equivalent since the early 2000s (Brun et al., 2017; Kääb et al., 2018). Positive mass-balances of the Aru glaciers (modeled based on ERA-interim reanalysis data; Kääb et al., 2018) and the widespread growth of endorheic lakes confirmed a precipitation increase in the region from the late 1990's on (Treichler et al., 2019). Driven by these positive mass balances, the two Aru glaciers underwent surge-like accelerations and mass transfers over several years before the detachments (Gilbert et al., 2018; Kääb et al., 2018). In apparent contrast to the positive mass balances and the surge-like mass transfers, the Aru-1 and Aru-2 glaciers both retreated by around 500 m between 1970 and 2015.

Since at least 2011 and until 2014, the sections above the eventual detachment zones of both Aru glaciers subsided. Simultaneously, glacier sections below bulged upwards. The rates of elevation change derived for 2011–2014 indicate that a down-glacier mass transfer had already begun during the second half of the 2000s (Gilbert et al., 2018). A modelling study based on the observed elevation changes reconstructed changes in basal friction and horizontal velocity prior to the detachments (Gilbert et al., 2018). It showed that the two glaciers were close to their steady state geometry with no/little sliding until 2010. Thereafter, decreasing friction under the whole detachment area of Aru-1 and in more localized zones of Aru-2 started to trigger the surge-like mass transfer. Modelling the glaciers' thermal regimes revealed that the frictional changes likely occurred in temperate areas of the two glaciers and that stress concentration occurred at the cold-ice margins (Gilbert et al., 2018). The surge-like changes of basal friction under the Aru glaciers were thus likely not associated with a change of the glaciers' thermal regimes but rather with a change in friction due to increasing water pressure in the already temperate areas. During the instability development, basal shear stresses in the detachment area dropped by an order of magnitude, leading to significant stress concentrations at the detachment margins and in a few spots under the glaciers. These stress concentrations led to strongly enhanced crevassing at the glacier margins and the zone of the later scarp head several months prior to the Aru-1 detachments. Fast-developing crevasses appeared only three weeks before the Aru-2 detachment and were discovered in satellite images in time to alert Chinese authorities.

The Aru glaciers are surrounded by continuous permafrost of -3 to -4 °C mean annual ground temperature (Obu et al., 2019). Field observations in the detachment and runout zones showed no presence of a hard-bed lithology beneath the glaciers, and very few large boulders were observed in the runout paths (own field visit and Lei et al. 2021). Rather, extensive deposits of soft, unconsolidated and fine-grained lithologies were identified. The Aru glaciers, situated in a region of positive or zero mass balances (Brun et al., 2017; Treichler et al., 2019) could well build up again to a size similar to the one before their 2016

collapses. Especially in the path of the Aru-1 avalanche, streamlined debris stripes, not present before the event, are well visible at several locations in high-resolution satellite data (Supplementary Fig. S5). We find no other evidence for a similar event in the recent past at the site, so finding two similar events at neighbouring glaciers is rather remarkable.

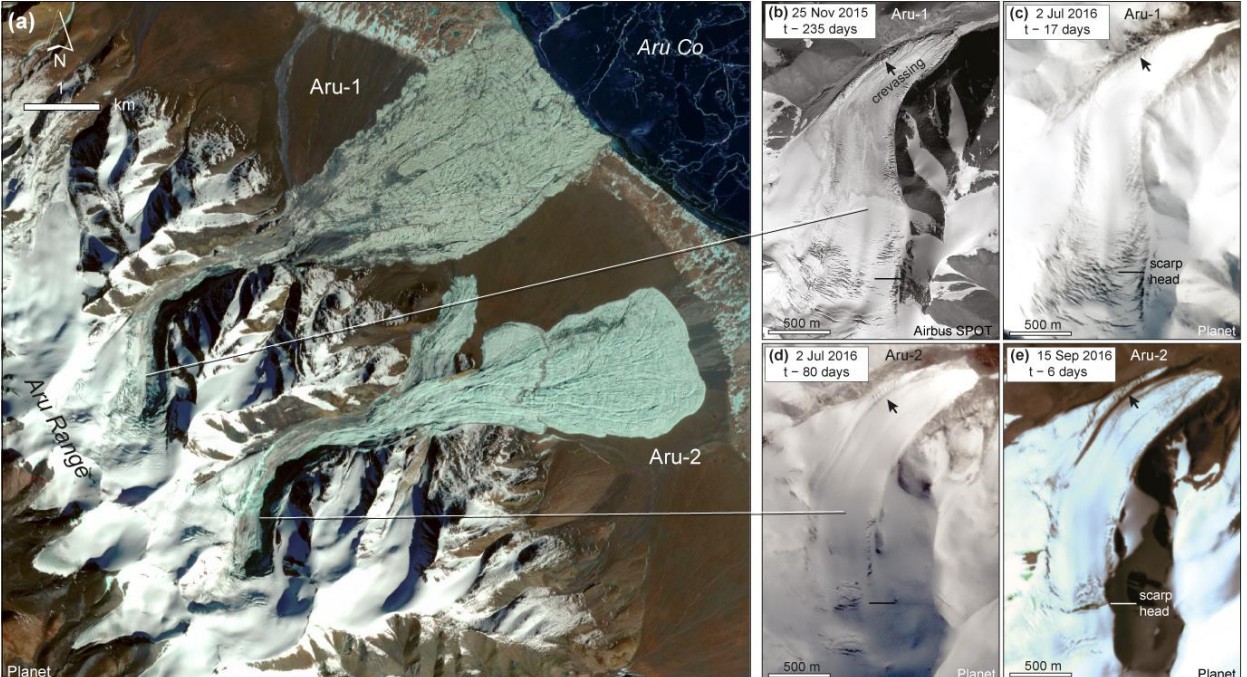

**Figure 9: Satellite images of the Aru glaciers, western Tibet, and their detachments. (a) Planet infra-red false-colour satellite image of 29 November 2016, after both glaciers collapsed (© Planet). (b) Enhanced crevassing on Aru-1 glacier (SPOT7, © Airbus). Time *t* refers to detachment date (17 July 2016 for Aru-1, 21 September 2016 for Aru-2). Note the particular crevassing at the northern curve of the glacier. (c), Aru-1 (Planet). (d), Aru-2 (SPOT7, © Airbus). (e), Aru-2 (© Planet), six days before collapse. The horizontal line in panels (b-e) indicates the scarp head positions of the later detachments.**

## 3.4 Tsambagarav, 1988, Altai, western Mongolia

Information about the 1988 Tsambagarav event is mainly relying on observations and interpretations by Avdeev et al. (1989). On the evening of 9 Aug 1988 (local time) the lower part of an unnamed glacier on the southern flank of Tsambagarav mountain (4193 m asl.), Altai mountains, western Mongolia, detached and formed an ice-rock avalanche. The avalanche travelled about 5.5 km (Avdeev et al., 1989), and from Landsat data we suggest the last 1-2 km might have been a mud flow (Fig. 10). As a peculiarity of this event, it seems to have been preconditioned by the 23 July 1988 Tsambagarav M6.4 earthquake. As a consequence of this earthquake, a large block (ca. $6 \cdot 10^6$ m$^3$) of the lower part of the glacier was separated from its upper parts

and displaced a few meters to south eastern direction. Melt water could then reach the glacier bed beneath the ice block through the developing crack. On 9 Aug 1988, the ice block detached from the glacier bed. Avdeev et al. (1989) describe the path of the resulting ice-rock avalanche in detail, but here we want to highlight two aspects of it. First, after about 1.8 km, the avalanche jumped (obviously at high speeds) over a 70-m tall ridge, leaving the ground on leeward side of the ridge intact. Second, Avdeev et al. (1989) estimate the deposited avalanche volume to about $12 \cdot 10^6$ m$^3$ (vs. $6 \cdot 10^6$ m$^3$ detached ice volume) indicating that the detachment must have eroded and ingested substantial amount of material along the way. The large debris content of the deposits likely also played a role in that it took almost 30 yr for the ice content to melt out completely (Agatova et al., 2020).

On a Corona satellite image from 11 Aug 1968, two obvious transverse crevasses, surrounded by a number of smaller concentric crevasses can be seen at the location of the upper scarp of the 1988 earthquake-triggered rupture and later glacier detachment (Fig. 10c). The entire feature has a diameter in the order of 150 m. Close visual inspection leads us to interpret the feature as a depression in the glacier surface. From many contemporary Planet satellite images with snow cover, and from Landsat 8 thermal data, we do not find indications of enhanced geothermal activity at the depression location. Today, the location appears to be the place for a spring that concentrates the subsurface runoff of the entire cirque. This spring could already have existed in 1988 and led to reduced basal friction and enhanced basal melt at its position, and thus to a depression on the glacier surface. Also today's glacier remains show a few obvious crevasses in the same area. Overall, the 1988 detachment seems to have happened at a pre-existing weakness and strongly crevassed location of the glacier.

Avdeev et al. (1989) note a slope angle of the detachment area of 20°-25°. From the SRTM1 DEM we find a slope of the valley bottom of around 15° at the location of the detached glacier part and assign an arbitrary 20° slope to the pre-detachment glacier. Since the detachment in 1988, the entire glacier has shrank substantially and as of 2020 only a small part of it is left. The detachment site lies in continuous permafrost with mean annual ground temperatures in the order of -10 °C (Obu et al., 2019). Avdeev et al. mention the existence of shattered bedrock in and around the glacier bed, and this is in line with visible interpretation of contemporary high-resolution satellite images. Ultimately, we cannot draw conclusions about the existence of particularly fine-grained sediments at the detachment site. From the weak avalanche traces that are still visible in high-resolution satellite images, it would be difficult to recognize the site as a place of a former glacier detachment and ice-rock avalanche. However, with the information from Avdeev et al. available, one can still detect signs of the past avalanche such as debris stripes oriented in avalanche direction and debris deposits inundating the mountain grassland (Fig. 10b). We do not find similar signs in other glacier valleys surrounding Tsambagarav, and the Corona spy satellite image from 11 Aug 1968 suggests that nothing like this has happened here before. Lastly, we are not aware of recent surge-type activity of glaciers in the Altai.

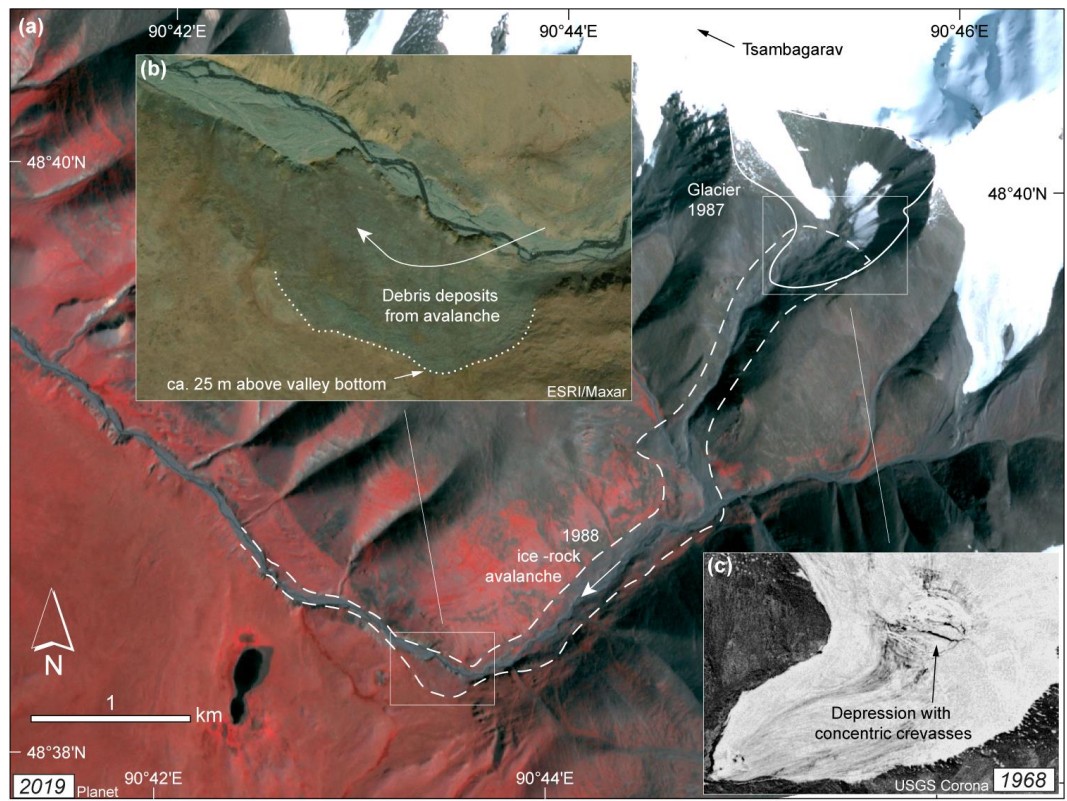

**Figure 10: (a) Location of the 9 Aug. 1988 Tsambagarav glacier detachment. Glacier outline of 1987 from Landsat TM**
**indicated as bold line, rough outline of ice-rock avalanche path from 1989 Landsat data as dashed line. The small white**
**rectangle indicates the position of panel (b) (Satellite image, 2019, © Planet). (b) Detail of 2015 Worldview data (©**
**ESRI/Maxar). The upper elevation of debris deposits from the avalanche in this right turn is about 25 m above valley**
**bottom. (c) A 11 Aug 1968 photo from a Corona satellite shows crevasses around a depression on the glacier surface,**
**at the location of the upper scarp of the 1988 detachment (credit: USGS).**

## 3.5 Amney Machen, 2004, 2007, 2016, and 2019, eastern Tibet

A sequence of surge-like advances, some of them ending in detachments of the glacier tongue, have been observed for a glacier
in the Amney Machen mountain range, Eastern Tibet (Fig. 11, Tab. 1; Paul, 2019). The isolated mountain range is home to
several surging glaciers (Wenying, 1983). The first ice-rock avalanche happened between 26 January and 3 February 2004,
and the involved volume was estimated to be $20\text{-}25 \cdot 10^6$ m$^3$, perhaps up to $36 \cdot 10^6$ m$^3$ according to a local information sign-
board (Paul, 2019). Three years later, between 23 September and 2 November 2007, a second detachment followed a surge-
460 like recovery of the glacier tongue that had detached in 2004. The 2007 detachment was considerably smaller in volume than
the one in 2004. A third detachment, also smaller than that in 2004, occurred between 4 and 7 October 2016 (date from Planet
images). Following this event, the glacier tongue started to recover again, and another small avalanche occurred from it

between 9 and 20 July 2019. Paul (2019) notes the weak rocks and fine sediments visible in the rock ribs in the glacier's steep source area. The glacier recovered rapidly after each detachment, suggesting that it is largely nourished by ice and rock fall from the headwall, and that a rock/ice melange likely makes up the glacier tongue. The surface slope of the detaching lower part of the glacier is around 15°. The glacier lies in an area of continuous permafrost with ground temperatures in the order of -3 to -6°C (Obu et al., 2019). Debris stripes oriented in avalanche direction are visible in high-resolution satellite images (not shown).

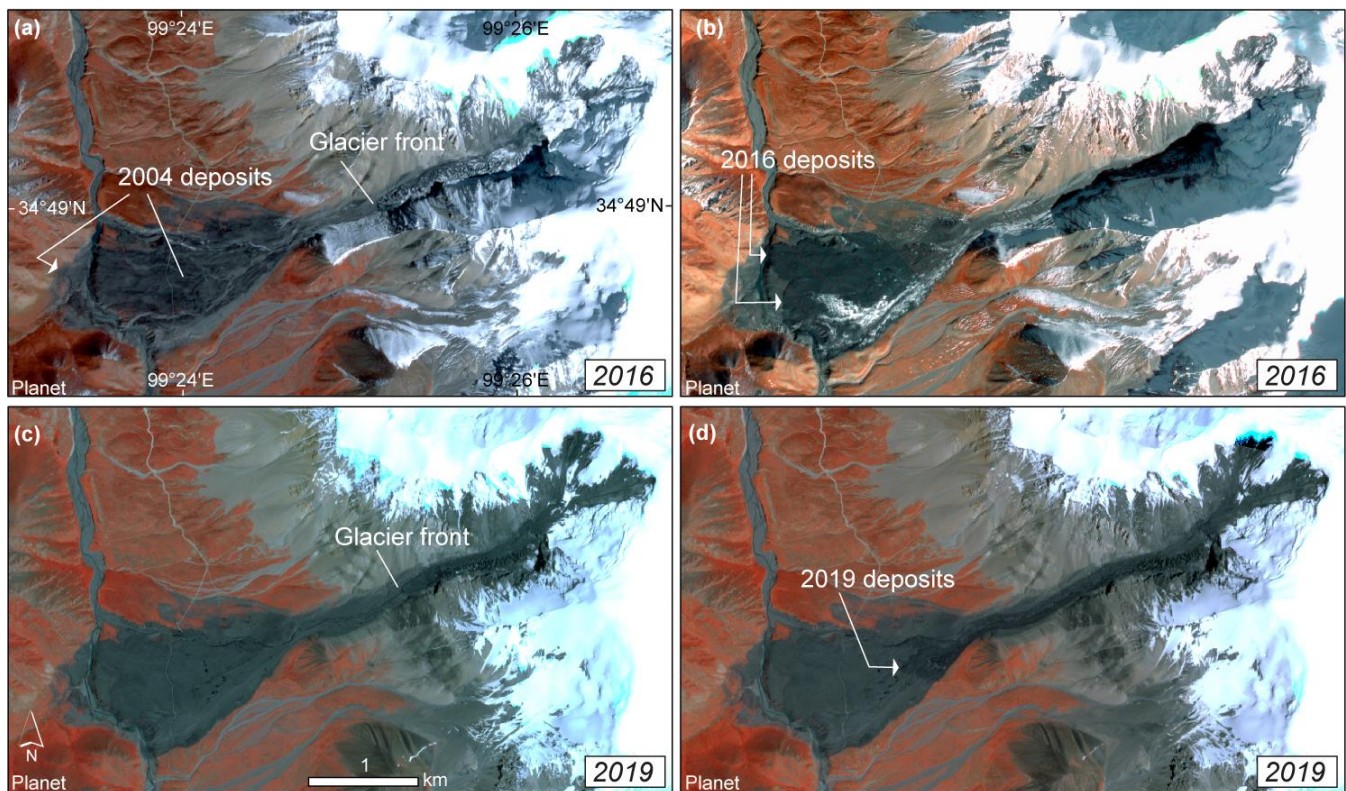

**Figure 11: Unnamed glacier in the Amney Machen range, Tibet. Satellite images of (a) 2 Oct 2016, (b) 23 Oct 2016, (c) 9 Jul 2019, and (d) 25 Jul 2019. Detachments of the glacier tongue happened before (a), between (a) and (b), and between (c) and (d). (Satellite images © Planet).**

## 3.6 Sedongpu/Gyala, 2018, south-eastern Tibet

During 2017 and 2018 the Sendongpu basin below the western flank of the Gyala Peri peak (7294 m.a.s.l.; Fig. 12) in south-eastern Tibet was the source of a series of large mass flows. Some of them dammed the Yarlung Tsangpo river, which posed a serious flood hazard to the upstream Gyalha village and large downstream areas, and triggered hazard management and

investigations of the causes (Tong et al., 2018; Liu et al., 2019; Chen et al., 2020). The largest of the mass flows stemmed from the detachment of a large, low-angle glacier in 2018.

### 3.6.1 Events before 2018

In order to elucidate significant elements of the recent mass-flow history from the basin that might have conditioned the 2018 detachment and that are not documented in detail elsewhere, we start our description with the oldest satellite data available to us. In Corona satellite reconnaissance data of 8 Dec 1969, a narrowing in the Yarlung Tsangpo river where the Sendongpu valley joins the river, points to deposits from previous mass flows. However, trees on these deposits suggest no recent large mass flow activity. The main glacier in the basin, here called Sedongpu Glacier (Fig. 12), shows some signs of enhanced flow such as large crevasses. On Corona satellite data of 6 Nov 1974 the glacier has advanced some 800 m into steeper terrain. Fresh traces of a large mass flow are visible between the glacier front and the main Yarlung Tsangpo river, and fresh deposits seem to have covered or destroyed the forest on the older deposits in the river, but the main glacier is still in place. The glacier showed one single tongue in 1969, but had split into two tongues during its advance by 1974, a feature that it still exhibited in 2016. Chen et al. (2020) describe an ice avalanche from the Sedongpu basin that dammed the Yarlung Tsangpo river in 1968, but do not mention any other events before 2014. From our interpretation of the Corona satellite data there was either one more event between 1969 and 1974, or the event dated '1968' actually happened a few years later. The next large mass flow (listed as ice avalanche in Chen et al. (2020) and debris flow in Tong et al. (2018); we interpret at least a large debris content) happened in 2014. The source of the mass flow must have been in the upper part of the main Sedongpu Glacier or its headwall, as both lateral moraines of the glacier were heavily eroded from the glacier side (RapidEye satellite data of 2013–2015). The glacier surface was not visibly changed along the glacier centre line, suggesting that the mass flow must have flowed along both glacier margins. Although difficult to determine in the satellite data available to us, the flow may have eroded the lowermost part of the glacier tongue.

Between 20 and 27 Oct 2017 (Planet and Sentinel-2 images) a huge rock avalanche started high up from the north ridge of Gyala Peri and ran over large parts of the Sedongpu basin and down to Yarlung Tsangpo, damming the river (Figs. 12 and 13). The event seems to have had a severe impact on Sedongpu Glacier. We generated two elevation models from 13 Nov 2015 Spot6 and 30 Dec 2018 Pleiades tri-stereo data that produced robust results despite the extreme topographic conditions. Differencing the two DEMs indicates that the October 2017 rock avalanche removed around 17 and $33 \cdot 10^6$ m$^3$ of material from two close-by but separated areas, respectively (Fig. 12c). If both failures happened as part of the same event, the total volume of $50 \cdot 10^6$ m$^3$ makes this one of the larger rock avalanches detected in recent decades. Based on visual inspection of satellite data, we consider it very likely that the avalanche also involved small glaciers from the west wall of Gyala Peri and incorporated ice from the surface of the glaciers lower down, as it ran over them. The Chinese seismic database registered two large 'landslide' events on 22 Oct 2017: A M3.2 event at 6:20 about 16 km west of Sedongpu and a M4.0 event at 6:22 directly at Sedongpu. Chen et al. (2020) and Tong et al. (2018) confirm that at least the latter signal stems from the Gyala Peri rock-

ice avalanche. Subsequent satellite images suggest that the avalanche must have changed the surface of Sedongpu Glacier drastically. It covered the glacier and much of the basin with debris and dust. A small, surge-like lobe with ponds on it appeared at the transition between the headwall and the tongue of Sedongpu Glacier (Fig. 13, and GoogleEarth, BingMaps). The eastern tributary glacier to Sedongpu Glacier also showed a surge-like lobe (Sentinel-2, Planet, GoogleEarth). Driven by the geomorphological changes in the basin, a series of debris flows, some involving ice and likely nourished from the large amounts of unconsolidated debris left behind by the rock-ice avalanche, occurred after 22 Oct 2017 and into 2018 (Tong et al., 2018; Chen et al., 2020). The two 18 Nov 2017 M5.2 and M6.9 Linzhi/Milin/Nyingchi earthquakes, with epicentres only a few km from Sedongpu, may have contributed to triggering of the debris flows (Hu et al., 2019). Lastly, we report hundreds of small earthquakes recorded under the Gyala Peri massif in 2017 and 2018, most up to M2, some up to M3, which we have not analysed further in the present study (Chinese Earthquake Data Center, 2020).

### 3.6.2 Detachment in 2018

Following the 2017 rock-ice avalanche the main Sedongpu Glacier underwent drastic changes (Fig. 13). Ponds developed on its surface and along the margins. Surface velocities increased from a background velocity of ~0.3 m/d (ca. 100 m/a) in 2017 to 1–3 m/d end of January 2018, 10 m/d in mid-September 2018, and 25 m/d in mid-October 2018 (velocities derived from offset tracking in repeat Planet, Sentinel-1, and Sentinel-2 data). The glacier surface showed several crevassed bulges (Fig. 13b) in January 2018, and progressivley more crevasses appeared as the glacier tongue expanded (Fig. 13c). The lower, flat glacier part separated from the steep head wall. Between 19 Sep 2018 (last optical image due to later cloud cover) and 13 Oct 2018 (Sentinel-1) the glacier advanced by almost 1 km. On 17 or 18 October (Tong et al., 2018; Chen et al., 2020), the entire tongue of Sedongpu Glacier detached over a length of about 3.5 km (glacier width between 250 and 550 m) (Figs. 12 and 13). Planet images of 27 Oct 2018 and Chinese media images confirm that large amounts of ice blocked the Yarlung Tsangpo river (Fig. 12d). Parts of the emptied glacier bed filled up with enough ice debris that another mass flow originated from there on 29 Oct 2018. The dam in the Yarlung Tsangpo river was estimated to be roughly $40–60 \cdot 10^6$ m$^3$ in volume (Chen et al., 2020). Differencing our 2015 SPOT6 and 30 Dec 2018 Pleiades tri-stereo DEMs (Fig. 13d) shows two areas of distinct volume loss: around $80 \cdot 10^6$ m$^3$ are missing from the main branch of the glacier, and around $50 \cdot 10^6$ m$^3$ from its terminus and frontal moraine. An ASTER satellite stereo DEM of 11 Nov 2017 suggests that the volume loss over the glacier tongue and frontal moraine cannot have occurred before November 2017. From our data we cannot tell how much of the total $130 \cdot 10^6$ m$^3$ stems from the main first glacier detachment event of 17/18 Oct 2018, and how much from the 29 Oct 2018 event. However, satellite images indicate that the first event involved by far the largest volume. The detached glacier part had an overall slope of only 8–9°. According to Obu et al. (2019) it should have been several hundred meters below the regional permafrost limit, but cold or polythermal ice is certainly found in the west wall of Gyala Peri, and could theoretically be advected into the basal parts of the glacier near its tongue.

Between 19 Sep and 26 Oct 2018 (Planet) a ~9·10⁶ m³ rock(-ice) avalanche (volume from 2015 SPOT6 and 2018 Pleiades tri-stereo DEM difference) originated from the south-western flank of Gyala Peri and likely reached Sedongpu Glacier (Fig. 12). The avalanche covered a small glacier below its starting zone which started a surge-like advance early 2018. Due to insufficient satellite data (cloud cover in optical data; low resolution and radar shadow in SAR data), though, we cannot tell if this avalanche happened before, during or after the 17/18 Oct 2018 glacier detachment, and could thus have triggered the glacier detachment. Seismic records are also inconclusive regarding the rock avalanche and the glacier detachment, with two dozens of M1–2 earthquakes recorded under the Gyala Peri massif between mid September and end of October 2018.

From all the evidence collected above, it seems very likely that the 22 Oct 2017 Gyala Peri rock avalanche, which travelled over the Sedongpu Glacier, primed the glacier for its detachment a year later, perhaps with additional influence of the 18 Nov 2017 earthquakes, though the exact controlling mechanisms remain unclear. The effects that the ongoing mass-wasting activities from Gyala Peri had on Sedongpu Glacier can have been manyfold: additional loading on the glacier can have increased normal and shear stresses, and the 2017 rock avalanche and earthquakes could have mechanically weakened the glacier and its bed, potentially disrupting the subglacial drainage system. The large amounts of fine dust deposited on the glacier will likely have changed (enhanced?) its surface melt rates. Lastly, independent factors such as high temperatures and precipitation amounts (Tong et al., 2018; Liu et al., 2019), associated with large water input into the glacier, could have complicated the changes caused by rock-avalanche impact. The supraglacial and glacier-marginal ponds, unusual on temperate glaciers (see Haeberli et al. 2002, Kääb et al. 2004), suggest that the glacier was under high internal water pressure. The bulging ice and the strong downstream gradient in surface velocities suggest that a surge-like instability first developed in the upper section of the low-angle part of the glacier. Just how this instability propagated down-glacier, whether by exerting pressure on the lower glacier from above, or by a propagation of exceptionally low basal friction values (Thogersen et al., 2019) remains unclear. High-resolution imagery and media photos of the glacier bed and detachment deposits, as well as the various debris flows that originated from the basin, suggest that the detached glacier rested on a soft bed with substantial amounts of fine material. The geology of the area is described as marble (Liu et al., 2019), which at least opens up the possibility of fine-grained sediments.

It remains to be seen to which extent the Sedongpu Glacier is able to rebuild, given the strongly negative mass balances in the region (Kääb et al., 2015; Brun et al., 2017; Treichler et al., 2019, Shean et al., 2020).

### 3.6.3 Zelunglung Glacier surge-like instabilities

The region around Sedongpu does not seem to host any obvious surge-type glaciers. However, the events at Zelunglung Glacier (29.62° N, 95.00° E; GLIMS G095018E29637N, RGI60-13.01428), 20 km south of Sedongpu, are worth mentioning. In 1950, 1968, and 1984 extraordinary instabilities propelled the glacier forward and blocked the Yarlung Tsangpo river (Zhang, 1992). The 1984 event seems to have involved only a smaller section of the glacier. Corona reconnaissance satellite images of 1969 show a massive advance (by about 4.5 km compared to 2018), but no detached glacier. However, the glacier had obviously

overridden its frontal moraine, reaching almost down to Yarlung Tsangpo. Deposits, visually similar to those of ice-rock avalanches in general cover much of the main Yarlung Tsanpgo river bed on a length of about 2.5 km downstream measured from the confluence with the Zelunglung valley. Glacier advance rates of up to 1 km/h are reported for the 1950 event (Zhang, 1992). Such rates are far above what is typical for surges and the glacier might have, at least in 1950, undergone an event close to a sudden detachment in the sense of the present contribution.

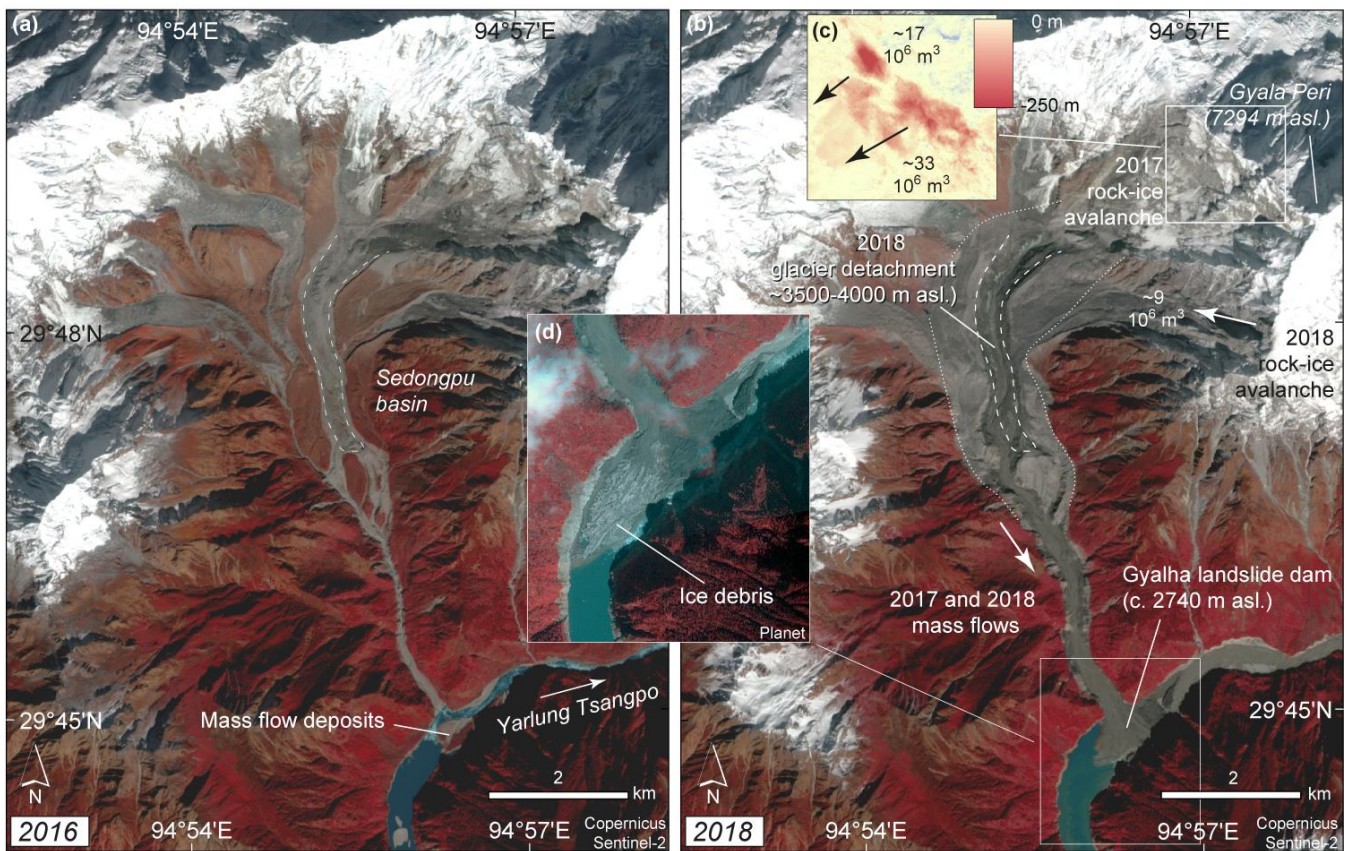

**Figure 12: Location of the 17/18 Oct 2018 Sedongpu glacier detachment, south-eastern Tibet. (a) Sentinel-2 image of 20 Nov 2016 (credit: Copernicus Sentinel data), before a series of mass flows happened that culminated in glacier detachment. Already earlier, mass flows from the basin have blocked the Yarlung Tsangpo river. (b) Sentinel-2 image of 31 Oct 2018 showing the area impacted by the 22 Oct 2017 rock avalanche from the Gyala Peri peak, and the 17/18 Oct 2018 glacier detachment. (c) Detail of elevation model differences between SPOT6 13 Nov 2015 and Pleiades 30 Dec 2018 tri-stereo data. Elevation losses amount up to 330 m, but the colour scale is saturated at -250 m. (d) Detail of the glacier detachment deposits in the main river (© Planet; image of 27 Oct 2018, 10 days after detachment).**

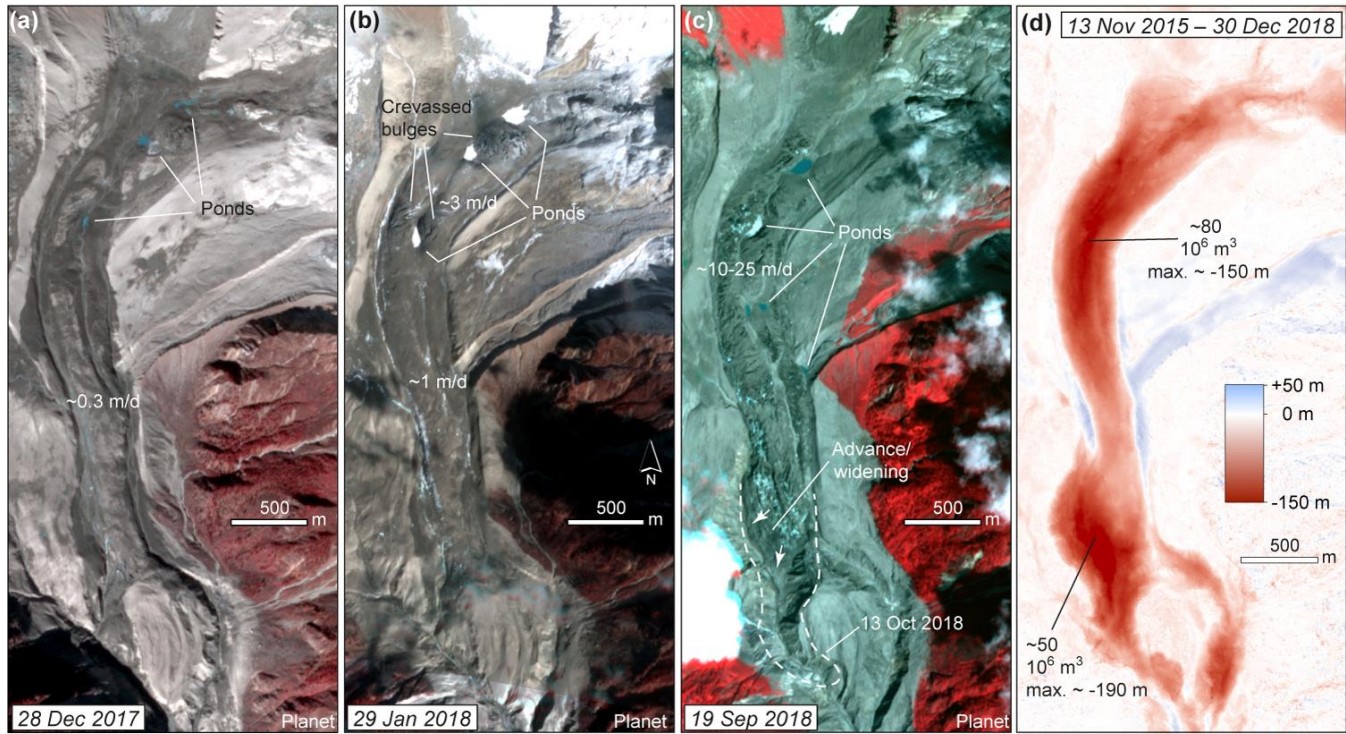

Figure 13: (a)-(c) Evolution of Sedongpu Glacier towards instability and detachment. Average surface velocities around the image dates are indicated. (Satellite images © Planet). Velocities and glacier position after 19 Sep 2018 (25 m/d) are derived from Sentinel-1 radar images. (d) Detail of elevation model differences between SPOT6 13 Nov 2015 and Pleiades 30 Dec 2018 tri-stereo data. Elevation losses amount up to 190 m, but the colour scale is saturated at -150 m.

## 3.7 Flat Creek, 2013, 2015 and 2016, Alaska

Flat Creek Glacier, a small glacier in the north-eastern corner of Alaska's St. Elias mountains (Fig. 14), produced two large glacier detachments in 2013 and 2015. This region in Alaska is home to many surging glaciers (e.g., Harrison et al., 2015; Sevestre and Benn, 2015; Kochtitzky et al., 2019), and a glacier in a valley adjacent to Flat Creek surged between 2012 and 2016. Located in the rain shadow of the St. Elias range, the area receives on average about 350 mm of precipitation annually (2008 – 2018). A mean annual air temperature of -14°C at the former terminus of Flat Creek Glacier, ground temperature measurements, and electrical resistivity tomography surveys strongly suggest that the headwall is underlain by continuous permafrost (Jacquemart et al., 2020).

On 5 August 2013, the lower 500 m of the glacier detached, releasing 6.8-11.2·$10^6$ $m^3$ of ice and lithic material. On 31 July 2015, most of the remaining glacier ice (up to the drainage divide) detached, evacuating an additional 17.6-20.1·$10^6$ $m^3$ (Jacquemart et al., 2020). Both events produced runouts of over 11 km (angle of reach 6-7°), deposited vast amounts of lithic

material, and buried several km$^2$ of old growth forest (400+ years old). The large amount of fine-grained sediment found in the deposits suggests that the failures occurred within the glacier bed rather than at the ice-bed interface. The detachment slope was determined to be ~20°.

A remarkable feature of Flat Creek glacier was a ~70 m tall bulge upstream of a stagnant, crevasse-free tongue. A similar bulge on Trapridge glacier, a polythermal surge-type glacier 80 km south-east of Flat Creek, was shown to have formed because a cold-ice tongue buttressed temperate ice upstream (Clarke and Blake, 1991). Based on the low annual air temperature and the presence of continuous permafrost, Jacquemart et al. (2020) concluded that the bulge on Flat Creek glacier was also the consequence of a polythermal regime. Measurements of the bulge position in satellite images suggest that the bulge advanced between 2011 and 2013, likely increasing driving stresses locally.

The 2013 and 2015 detachments both occurred at the peak of their respective melt seasons. Using a degree-day model, Jacquemart et al. (2020) found that the water availability during the exceptionally warm summer of 2013 was primarily melt driven and up to 4.8 standard deviations ($\sigma$) above the long term mean (1979–2015). No detachments were detected in 2014, when water availability was below average (-0.5 $\sigma$). Water availability was again higher in 2015 (+ 1 $\sigma$), when the second detachment occurred.

In 2016, a third glacier detachment released from a much steeper glacier (~30°) in the same cirque (not described in Jacquemart et al. (2020) but mentioned in Jacquemart and Loso (2019)). We differenced a 13 March 2016 Arctic DEM and a structure-from-motion DEM from a 2019 aerial survey and estimate the volume of this detachment to be $4.7 \pm 0.18 \cdot 10^6$ m$^3$. DEM differences over 2012–2016 show bulging of up to 30 m in the lower part of the later detachment and surface lowering in its upper part (Fig. 14 f, and Supplement). The detachment causes have not yet been investigated, but the moving mass of ice was caught on video by rangers on a coincidental overflight. The speed of the observed mass flow was about three times slower than that of the 2013 and 2015 events. Nevertheless, the churning mass of ice blocks (termed a "slush-avalanche" by Jacquemart and Loso (2019)) provides a sense of what the much larger flows may have looked like.

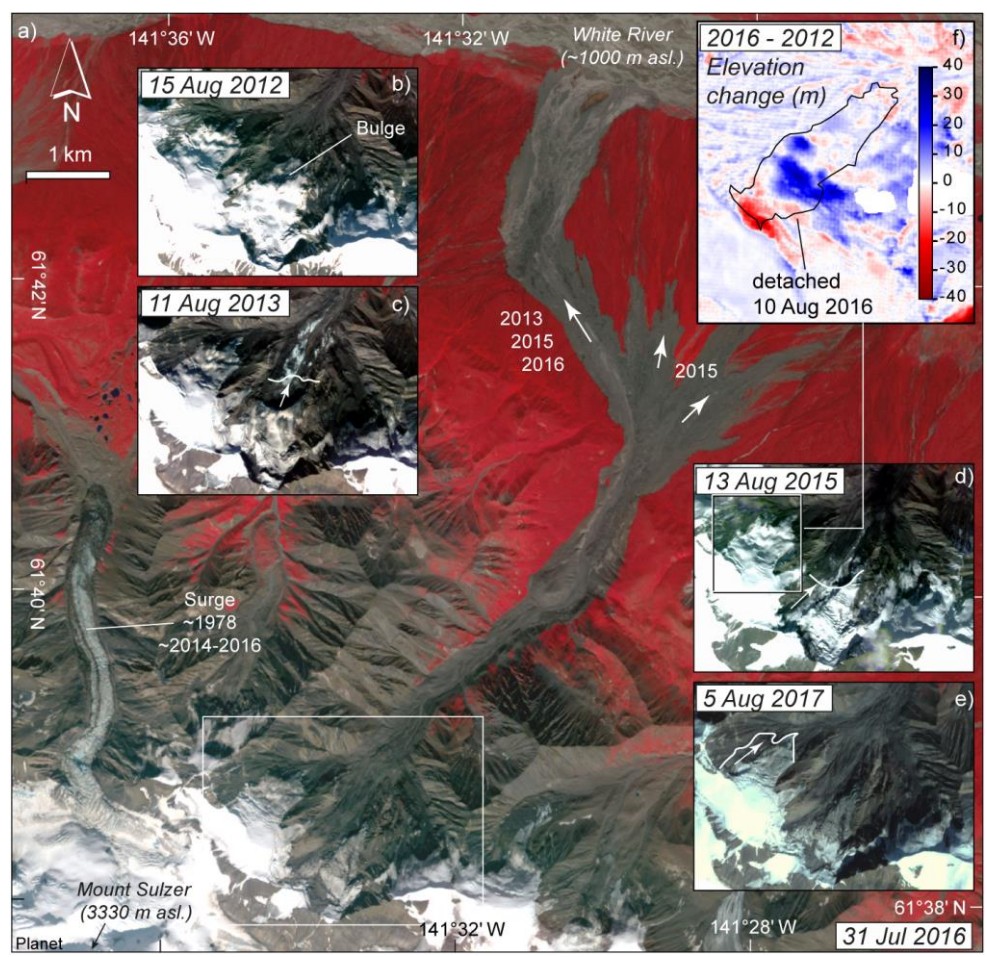

Figure 14: a) Overview of the three detachments and ice-rock avalanches from Flat Creek Glacier, Alaska, since 2012. b) On the 15 Aug 2012 inset, a bulge is visible on the lower part of the glacier. c) On the 11 Aug 2013 inset, the glacier part that detached on 5 Aug 2013 is indicated together with the front position before the event (white line). d) On the 13 Aug 2015 inset, the detachment of 31 Jul 2015 and the front position before it are indicated. e) On the 5 Aug 2017 inset, the area of the 10 Aug 2016 detachment is shown. All images © Planet (Dove and RapidEye satellites). (For more images and front positions see Jacquemart et al. (2020)). f) DEM difference between 13 Mar 2016 (Arctic DEM) and summer 2012 (Alaska IfSAR) showing surge-like mass redistribution (see also Supplement).

## 3.8 Aparejo, 1980, Chilean Andes

On 1 March 1980, $7 \cdot 10^6$ m$^3$ (85% of its total volume) sheared off the debris-covered Aparejo Glacier in the Chilean Andes, mobilizing the detached mass 3.7 km down-valley with an estimated speed of 110 km/h (Ugalde et al., 2015; Ugalde, 2016; Ugalde et al., 2017) (Fig. 15). The slide deposit covered an area of 0.55 km$^2$ with ice and rock debris piled up to 17 m thick. The volume of the deposit was estimated at $8.1 \cdot 10^6$ m$^3$ (Marangunic, 1980). Five mountaineers witnessed the event and noted

several supraglacial ponds and 2-3 cm of wet snow on the surface of the glacier (Marangunic, 1980). These observations suggest that the triggering mechanism of the glacier detachment likely involved an extreme reduction of the basal drag due to high water saturation of the glacier bed. Aparejo Glacier appears to sit on a glacier bed composed primarily of weak subglacial till , and the slope on the lower two thirds of the glacier averages 7°. Snowmelt infiltration and warm precipitation due to a sudden increase of the zero-degree isotherm elevation could have provided the main source of infiltrated water, leading to enhanced water pressure at the glacier bed. During a field inspection on 12 March 1980, Marangunic (1980) found that the nearby debris-covered glacier to the east, glacier no 51 according to the Chilean glacier inventory at the time (Fig. 15), also showed significant signs of surge-like instability, such as a heavily crevassed front and patches of freshly exposed ice along its entire length. The prominent terminal moraine of this glacier may have contained its detachment, though.

The Aparejo glacier is situated in a region of complex geology with a number of weak rock formations, including sandstones and fine-grained conglomerate in the immediate vicinity of the glacier (Ugalde, 2016). Ugalde (2016) sampled the grain size distribution of the remains of the lower ice-rock avalanche deposits and did not find them to contain more fines than a typical moraine, but notes that spatial variability was high and that the 35 years since the detachment mav have depleted the deposits of fine particles. In the former avalanche path, modern satellite images show streamlined debris stripes similar to those reported from several other detachments in this contribution. Remarkably, similar debris stripes are also visible in Hycon airphotos from 1956. Interestingly, the geomorphology of the deposit area is similar between the 1956 airphotos and the post 1980-event high-resolution satellite images. One possible interpretation of this is that large mass flows had already originated from the Aparejo cirque at earlier times. A detailed field investigation would be required to determine whether the debris stripes consist of glacial flutes formed under a previous glacier extent or stem from a catastrophic detachment.

In 2015, the glacier had around 15% of its pre-detachment volume, covering much of the original area, and with a surface slope of around 20° (Ugalde, 2016). The current glacier terminus lies at around 3400 m a.s.l., slightly below the lower regional limit of discontinuous permafrost, estimated by Brenning (2005) to be at around 3500 m a.s.l. Consistent with regional glacier mass balance trends (Falaschi et al., 2018b; Braun et al., 2019; Dussaillant et al., 2019), decreasing ice thicknesses have been identified on the lower and middle section of the Aparejo glacier (Ugalde et al., 2017), pointing to the glacier shrinking under current climate conditions.

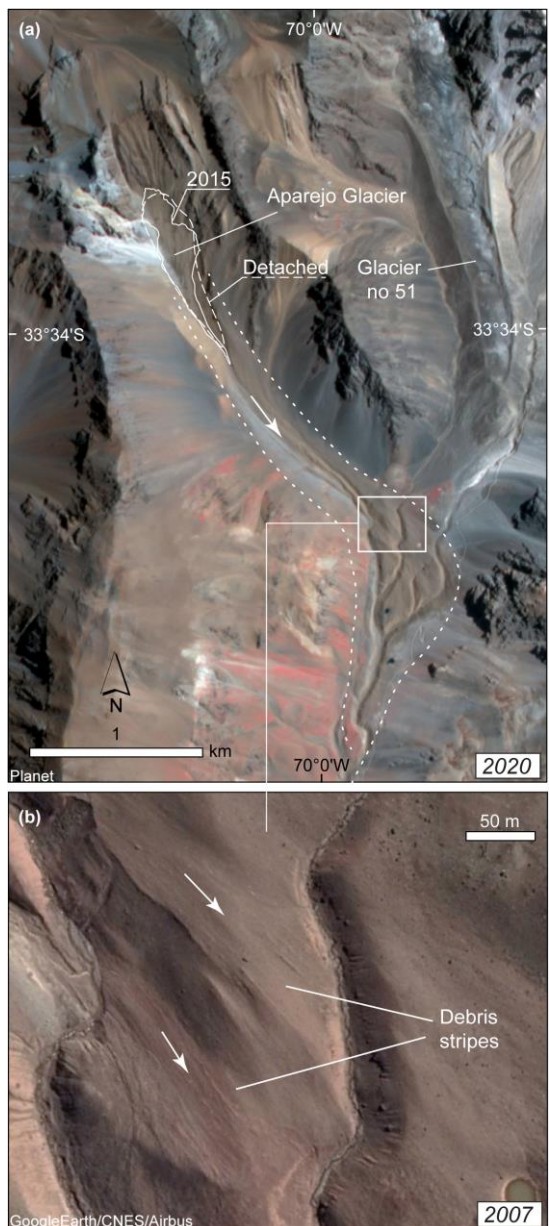

**Figure 15: Aparejo Glacier, Chilean Andes. (a) Satellite image (© Planet) of 27 Jan 2020 with the outlines of the detached glacier as solid line, and the outlines of the 1980 detachment and runout roughly indicated as dashed lines (outlines after Marangunic, 1980; Ugalde, 2016; Ugalde et al., 2017). (b) Detail of avalanche path (rectangle in (a)) with streamlined debris stripes. Satellite image: © Google Earth and CNES/Airbus, 29 Feb 2016.**

## 3.9 Leñas, 2007, Argentinean Andes

The $4 \cdot 10^6$ m$^3$ detachment of Leñas Glacier between 5 and 14 March 2007 was discovered only recently, since it happened in a very remote region and had no down-stream impacts (Fig. 16). Meanwhile, the case is described in detail by Falaschi et al. (2019). The detached lower glacier section had a surface slope of around 15-16°. The glacier tongue at around 3450 m a.s.l. is suggested to lie within the zone of discontinuous permafrost in the region. The bed characteristics of the glacier are not known, but large amounts of fine sediments were found over the run-out area of the event. A number of glacier surges are documented in the wider region but not for Leñas Glacier itself (Falaschi et al., 2018a, 2019). At the location of the later head scarp of the detachment, pronounced transverse crevasses are visible in airphotos from 1970 and in a satellite image from a few weeks before the event (SPOT5). It is hard to determine, however, if these crevasses could have been signs of abnormal glacier behaviour, or rather a pre-existing feature that then naturally formed the upper failure scarp. Transverse undulations of the glacier surface as indicated in the 2000 SRTM DEM, and a slight increase of surface gradients at the location of the crevasse zone favour the latter scenario.

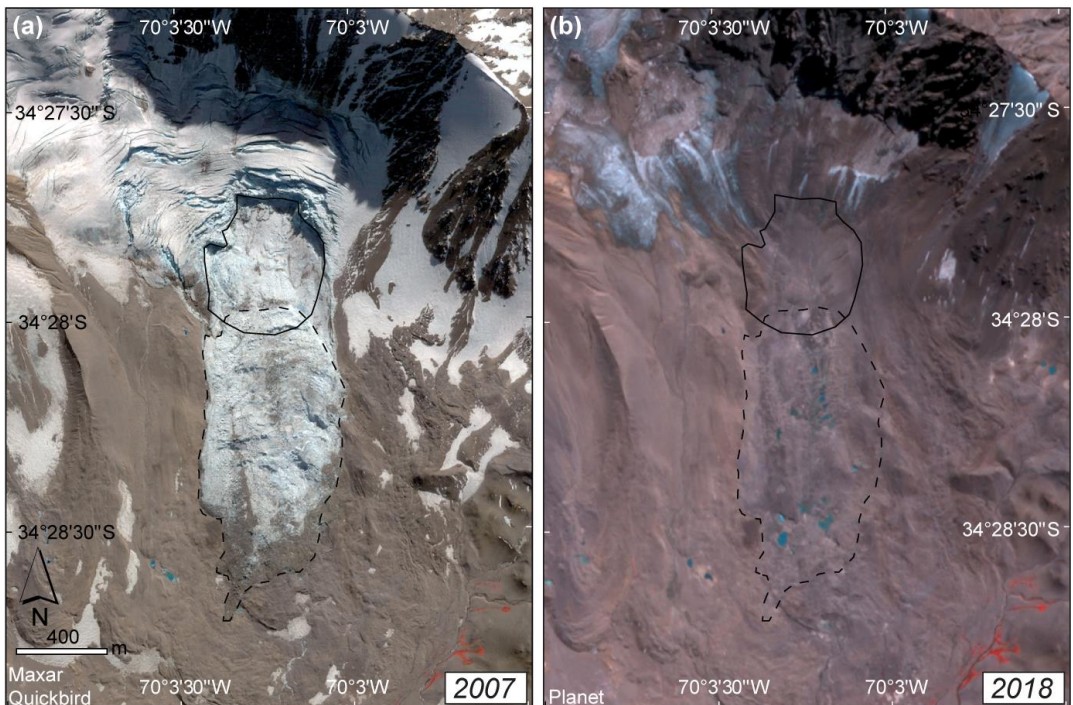

**Figure 16: (a) Quickbird satellite image of 19 Apr 2007 over Leñas Glacier, Argentinian Andes, a few days after detachment (© Maxar). (b) Planet image of 28 Mar 2018. Bold lines in both panels indicate the detached glacier part, dashed lines the outlines of the avalanche deposits. (© Planet).**

## 3.10 Tinguiririca, 1994 and 2007, Chilean Andes

### 3.10.1 The 1994, 2007 and a possible 1962 event

A detachment from a glacier on the southern flank of the Tinguiririca Volcano (Fig. 17) happened between 27 June (Landsat 5 TM) and 6 July 1994 (Landsat 5 TM). For the event, we estimate a detached glacier area of 0.2 km$^2$. Using glacier thickness estimates derived for the 2007 case (described below), we estimate a detachment volume of roughly 4–5·10$^6$ m$^3$. The climatic conditions between 1994 and 2007 were obviously favourable enough (i.e. little negative or even balanced regional glacier mass balances; Masiokas et al, 2016; Dussaillant et al. 2019) for the glacier to recover to its pre-detachment geometry.

At the same location, a glacier area of 0.46 km$^2$ detached between 7 January (Landsat ETM+ image without collapse) and 14 January 2007 (Landsat ETM+ with collapse), producing an ice/debris avalanche of 10–14·10$^6$ m$^3$ volume (Schneider et al., 2011; Iribarren Anacona et al., 2015; Figs. 17, 18). We estimate that the detached glacier had a surface slope of around 20°. Before the 2007 detachment, the glacier's lowest elevation was at about 3500 m a.s.l., roughly at the lower regional limit of discontinuous permafrost (Brenning, 2005). The volcanic nature of Tinguiririca should be associated with weak rocks and sediments. There are fumarolic fields and hot springs within a few km of the detached glacier (Pavez et al., 2016), and an inactive volcanic crater lies just a few hundred metres to the west of the detached glacier. In very high-resolution satellite images of 2007 and later (GoogleEarth, BingMaps), clear signs of hydrologic activity are visible in the upper part of the detachment area: freshly eroded channels, wet looking areas, and deposits from small debris flows – all perhaps signs of geothermally-enhanced melt of snow and ice (Fig. 18). In the path of the ice-rock avalanche we find streamlined debris stripes similar to those found in the Kolka, Rasht 2019, and Aparejo avalanche paths (Fig. 18). As of late 2019, the glacier has not recovered from its 2007 detachment and there are only a few small snow (or ice?) fields visible at its location. Elevation differences between 2000 (SRTM DEM, before detachment), 2007–2010 (ALOS PRISM, after detachment), and 2010–2015 (TanDEM-X; both the ALOS PRISM and TanDEM-X DEMs are multi-year composites and therefore their date range is given) suggest that the detached glacier had average and maximum thickness of 21–28 m and 50 m, respectively. Combined with the detached glacier area of about 0.5 km$^2$ we estimate a detachment volume of roughly 10.5–14·10$^6$ m$^3$ , which is in good agreement with Iribarren Anacona et al. (2015). The Tinguiririca case illustrates how glacier removal by the 1994 detachment was largely reversible as the glacier built up again to failure conditions under the climate at that time, whereas the 2007 glacier removal seems irreversible under current climate and associated negative glacier mass balances in the region (Falaschi et al., 2018b; Braun et al., 2019; Dussaillant et al., 2019).

In airphotos from 1962 and Corona-series reconnaissance-satellite images from 1967 we notice debris stripes in the valley similar to the ones visible after the 1994 and 2007 events, and other cases described in this contribution. Over the glacier, the 1962 airphotos clearly show that a glacier detachment similar to the 1994 event must have occurred not long (weeks, months or a few years?) before the image was taken. Stripes of debris or ice remains after the avalanche are still well preserved in the 1962 images over the glacier (Fig. 18d). (The associated terrain sections are under snow cover in the 1967 Corona images).

## 3.9.2 A potential large pre-1970s detachment of the neighbouring glacier

In the neighbouring valley to the east (Fig. 19) we note geomorphological traces that could be investigated further with regard to their origin from volcanic mass movements, earlier glacier stages, glacier surging, or more glacier-detachment like events.

In the lower part of this potential event path we find debris stripes, similar to the ones from the Tinguiririca, Aparejo and other avalanches of this study, largely unchanged since the first available Corona-series reconnaissance-satellite images in 1967

(Fig. 19c), and airphotos from 1962. Only detailed field work would be able to rule out the possibility that this debris stripes consist of glacial flutes or small lateral moraines formed during a previous glacier extent, or were produced by a glacier surge.

It should also be investigated whether such debris stripes can remain largely intact after having been overrun by an ice-rock avalanche or if it is an indicator of the most recent event (see also Aparejo, where similar questions turned up).

In the upper part of this valley, or in the potential event source area, we observe two noticeable changes over time. First, the tongue of this unnamed glacier (RGI60-17.01112, GLIMS ID: G289692E34781S) was much smaller in 1962 airphotos than it

is in 2020. Its advance of roughly 1.5 km since 1962, most of which occurred between 1975 and 1986 (Landsat), is in stark contrast to the pronounced shrinkage of the other glaciers in the area (Fig. 17 and 19). Indeed, differencing a DEM, which we

produced from 1962 stereo-airphotos, from the SRTM or TanDEM-X DEMs confirms elevation gains on the glacier tongue of up to 120 m between 1962–2000 (SRTM), or up to 150 m between 1962 and 2010-2015 (TanDEM-X; Fig. 19c). This

development could point to the recovery of the glacier tongue after a removal some time before 1962. The volume gain of the glacier tongue between 1962 and 2000 is around $70 \cdot 10^6$ m$^3$, or $100 \cdot 10^6$ m$^3$ between1962 and 2015.

Second, we detected an elevation decrease of the glacier forefield of around 10–15 m between 1962 and 2000, which could point to the deflation of debris-covered ground ice, deposited by a possible surge or glacier detachment (Fig. 19c). Third, a

comparison of the 1962 airphotos to contemporary images revealed a heavily bulged glacier surface between 3900 m a.s.l. and 4650 m a.s.l. (Fig. 19a,b). Visually, this bulging is similar to the bulge found on Flat Creek Glacier prior to its 2013 detachment

(section 3.7). The slope of the glacier tongue is around 8-10°, and around 35° for the steep upper part. The bulging upper part of the glacier is far above the regional permafrost limit so that polythermal ice conditions might well be found in parts of it.

We also examined a 1955 Hycon aerial photo (no stereo data to produce a DEM available to us), but visual interpretations from it remain inconclusive. Under illumination conditions that are very different from those of the 1962 images, the headwall

glacierisation and the tongue seem larger in 1955 than in 1962, resembling rather its shape and extent of the 1980s. Below the position of the current (2020) glacier terminus, there seem to be dead-ice remains visible in the 1955 images. These are also

visible in the 1962 images, though shrinked. It remains thus to be clarified at this point to which extent the features observed can be explained by a surge, or series of surges of the glacier before 1955, or between 1955 and 1962, or by a detachment-like

event.

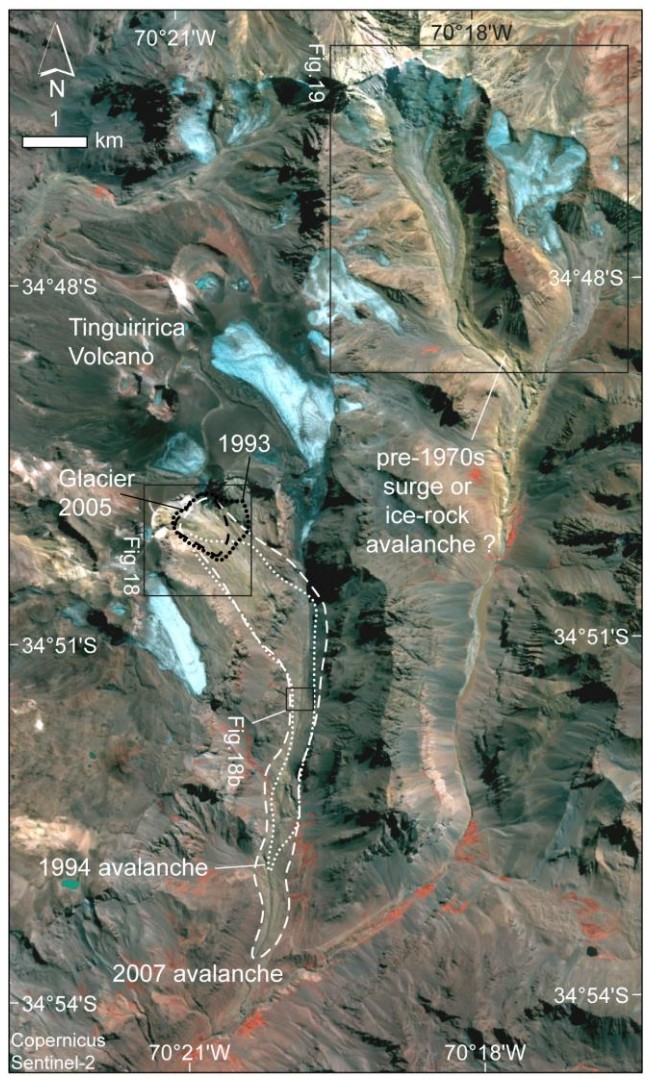

**Figure 17: Tinguiririca Volcano, Chilean Andes. Outlines of the 1994 and 2007 ice-rock avalanches, and the 1993 and 2005 area of the detached glacier as dotted or dashed lines, respectively. These outlines have been digitised for this study but were found to agree well with the ones in Iribarren Anacona et al. (2015). Satellite image: Sentinel-2, 14 Mar 2020 (credit: Copernicus Sentinel data).**

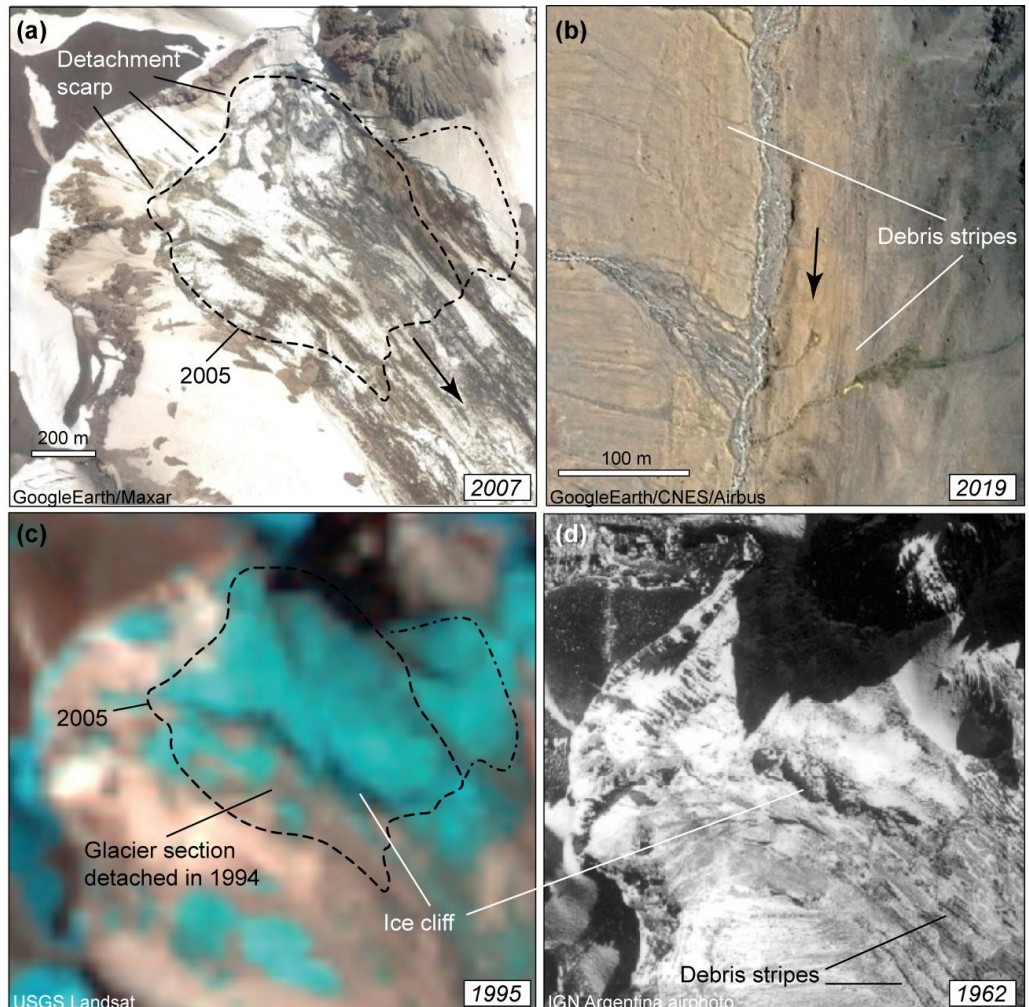

**Figure 18: (a) Glacier detachment area at Tinguiririca Volcano on 24 Jan 2007 (satellite image: © Google Earth and Maxar). See Fig. 17 for location. Glacier outlines from 26 Feb 2005 Landsat data (latest snow-free image before 2007 detachment) indicated by dashed line. From the 30-m resolution Landsat data it is unclear whether the dash-dotted ice section was connected to the main glacier. (b) Debris stripes left by the 2007 ice-rock avalanche (30 Jan 2019; © Google Earth and CNES/Airbus). (c) A Landsat image of 19 Mar 1995 (first snow-free image after 1994 detachment; credit: USGS) shows that the southern part of the glacier detached in the 1994 event. Glacier outlines of 2005 dashed as in (a). (d) The glacier on an aerial image of 8 Apr 1962 (© National Geographical Institute of Argentina). Clearly, a glacier section similar to the 1994 event has detached not long before the image date, leaving also a similar ice cliff. Streamlined debris stripes from the resulting avalanche are still well visible.**

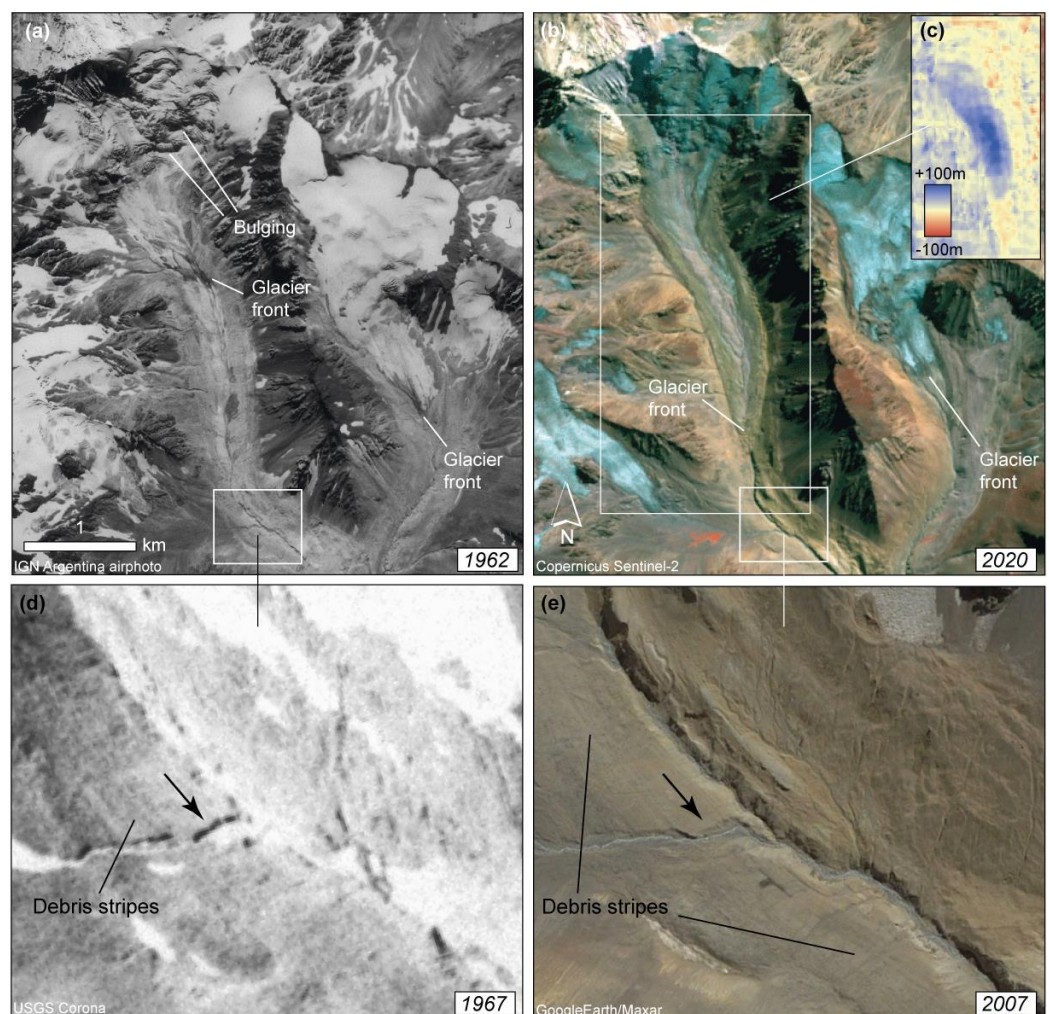

Figure 19: Glacier valley with potential former surge or glacier detachment, east of Tinguiririca Volcano. See Fig. 17 for location. (a) Orthoprojected airphoto of 8 Apr 1962 (© National Geographical Institute of Argentina). (b) Sentinel-2 image of 14 Mar 2020, showing an advance of 1.5 km relative to 1962, in contrast the other retreating glaciers in the area (credit: Copernicus Sentinel data). (c) Elevation differences between DEMs from 1962 stereo airphotos and the 2000 SRTM show gains over the glacier tongue of up to 120 m, and losses of around 10-15 m in the forefield. (d) Corona satellite image of 23 Feb 1967 with debris stripes (credit: USGS). (e) Satellite image of 19 Mar 2007 (© Google Earth and Maxar). (a) and (b), and (d) and (e) show the same terrain section each. Locations of (c), (d) and (e) indicated as white rectangles in (a) and (b).

# 4 Discussion

## 4.1 Simplified force and energy balance

### 4.1.1 Idealized slab model

The condition under which a detachment might occur can be qualitatively understood from a simple force balance analysis. To simplify the problem, we represent the potential detachment as a rectangular slab (Fig. 20a). The force balance between gravitational (left part of Eq. 1) and resistance forces (right part of Eq. 1) gives the following relationship:

$$W\,L\,h\,\rho\,g\,sin(\alpha) = W\,L\,\tau_b + 2\,h\,L\,\tau_d \qquad (1)$$

and by solving for $\tau_d$ :

$$\tau_d = \frac{W}{2h}(\rho g h \sin(\alpha) - \tau_b) \qquad (2)$$

where $\tau_d$ is the lateral shear stress (Pa), W is the slab width (m), L is the slab length (m; cancelling out), h is the slab thickness (m), $\rho$ the density of ice (kg m$^{-3}$), g is the gravitational acceleration (m s$^{-2}$), $\alpha$ is the slab slope, and $\tau_b$ is the basal shear stress (Pa). Assuming that the glacier driving stress is originally in balance with the basal shear stress under normal conditions ($\tau_b = \tau_0$) we have:

$$\tau_0 = \rho g h \, sin(\alpha) \qquad (3)$$

which gives the following equation by combining (2) and (3) in a way that a ratio $\frac{\tau_b}{\tau_0}$ appears, which allows us to compare potential failure conditions to normal conditions:

$$\tau_d = \frac{W \rho g \, \sin(\alpha)}{2}\left(1 - \frac{\tau_b}{\tau_0}\right) \qquad (4)$$

A collapse can only happen if $\tau_d$ exceeds the mechanical resistance of the slab margins (critical shear strength $\tau_c$). Using the work of Gilbert at al. (2018) where the evolution of the force balance toward the collapse of Aru glaciers has been quantified, we estimate $\tau_c \cong 0.28$ MPa for the Aru glaciers and apply this value to the simplified glacier slab. This allows, using Eq. (4), to define a stability diagram as a function of detachment width and slope, and the ratio $\frac{\tau_b}{\tau_0}$ (Fig. 20b).

This stability criterion seems to be respected by most of the detachments of our study (Fig. 20b) showing that the rough slab approximation may be a way to initially and qualitatively analyse which glaciers might be susceptible to detaching. In particular, the analysis shows which combinations of slope and width are unlikely to produce detachment when $\tau_d$ remains below $\tau_c$ even for $\tau_b = 0$ (i.e. complete loss of basal friction; green area in Fig 20b). Under the provisional assumption that $\tau_c$ is similar for all the glaciers investigated, this analysis also shows that the detachments presented here happened for an average loss of friction between around 50 and 100% (i.e. $0 < \tau_b/\tau_0 < 0.5$). In reality, it is reasonable to assume that the basal friction

rarely goes all the way to zero. Additionally, the critical lateral shear stress will vary between glaciers, depending for instance on whether the critical lateral resistance is provided by ice or by morainic margins, their properties, and on topographic forms

of resistance like glacier curves or bedrock bumps. For the Amney Machen detachment, which did not follow our provisional criteria (it failed at a critical shear stress of 0.2 MPa, which is smaller than the 0.28 MPa estimated for Aru), the detachment

flanks consisted of sediments and might thus be weaker than the ice margins along which the Aru glaciers detached. For some glaciers the effective width of the detachment is difficult to estimate as it varies along the glacier length and we cannot be sure

at which width the failure started to develop. For Tinguiririca, the effective detachment width is particularly uncertain as the glacier rested on a bed ramp rather than between valley or moraine flanks. It remains to be investigated how the deviation of

the cross-sections of the detached glaciers from the idealized slab influences our stability analysis.

### 4.1.2 Energy from precursory acceleration

As will be discussed in the following sections in more detail, several of the detachments presented here showed precursory accelerations, some clearly surge-like. In the present section we discuss to what extent the melt-water production associated

with this precursory motion could feedback on the reduction of basal shear stress. We take the case of the Aru-1 event (based on data in Gilbert et al. 2018) as an example. The first major dissipative losses of this event were during the pre-detachment

accelerated sliding, which attained about 0.5 m/day. Whereas some energy drove crevasse development and brittle-ductile changes in ice crystals, we suppose that most of this phase of energy dissipation occurred at the bed by grinding of rocks and

ice, and melting of ice. The typical glacier thickness was 100 m, so 1 $m^2$ of bed area by 100 m column of ice at a density of 900 kg/$m^3$ would have a mass of about 90,000 kg. Each day it slid downslope by about 0.5 m, including an elevation drop of

about 0.1m. The daily loss of potential energy of each such column of ice was thus about 88,300 J (about 1 W/$m^2$). If the ice was at the melting point, which it was across much of the bed within the frozen perimeter (Gilbert et al., 2018), this energy

could melt up to around 0.26 kg of ice per $m^2$ of thawed bed per day. If this continued for 200 days at such rapid sliding, about 53 kg of liquid water would be generated per $m^2$ of thawed bed area, amounting to a roughly 5 cm layer of water (some energy

may also have been expended in crushing rocks and ice). This water could be expelled, or it could be contained within the frozen confines of the polythermal glacier. If confined, this water could be ingested into basal till, perhaps dilating the volume

of till and spreading it over a larger fraction of the bed, thus reducing the total frictional resistance in the thawed parts of the glacier. Or the water itself could spread over a larger fraction of the bed, or it could pool against the frozen toe and margins of

the glacier. Additional water – maybe a lot more – likely was provided to the subglacial bed environment from rainfall and snowmelt sources. The role of the above 5-cm average thickness of frictional meltwater could become important if (i) it is a

substantial proportion of the thickness of the basal till, (ii) if it is a significant proportion of the meteorological meltwater that reaches the bed, and/or (iii) if it can spread laterally over a large part of the thawed bed.

Not all detachments presented here were preceded by longer phases of surge-like acceleration, and surge-like acceleration typically does not lead to glacier detachment. Still, the above estimates exhibit a feedback process that could through

precursory acceleration facilitate substantial reduction of basal shear stress and ultimately failure of soft beds.

## 4.2 Similarities and differences

The most apparent similarities among the detachments compiled in this contribution (Tab. 1) are their geographic proximity to surge-type glaciers – in some cases the detached glaciers themselves exhibited surging or surge-like behaviour – as well as

the existence of weak bedrock and/or fine sediments around and likely at the base of the glaciers (Fig. 21). Both commonalities suggest that the sudden detachments of low-angle glaciers discussed here could be seen as rare and extreme endmembers of

the range of surge-type and surge-like glacier instabilities (Quincey et al., 2015; Herreid and Truffer, 2016). The detachments could be a specific kind of glacier surge where the force balance cannot be achieved by a global control, typically by

longitudinal and lateral stresses, when basal friction is suddenly reduced. In case of a low bed roughness and an absence of sufficient topographic support, the reduced amount of stress accommodated by basal resistance can only be transferred to the

margins (Fig. 20a) and leads to an expanding instability (Thogersen et al., 2019). This ultimately leads to a runaway acceleration and detachment. The loss of friction involved in such behaviour, more than 50% according to our idealized slab

analysis, may only be reached by sustained low effective pressure. Such conditions are more plausible on soft-bed glaciers in contrast to hard-bed glaciers where increasing sliding velocity would lead to cavity opening and increasing drainage efficiency,

making preservation of high water pressure at the glacier bed unlikely.

In this context, we note that the surface slopes of the detached glaciers were between 9° and 21° (average 15.9°, standard

deviation 3.6°), which is at the upper end for slopes of surge-type glaciers, yet surprisingly low for glaciers causing ice avalanches. The slope range of around 10–20° may be a necessary condition for glacier detachments, as a slope lower than 9°

is unlikely to drive a stress concentration that exceeds the critical shear stress even in the case of total loss of basal shear stress (Figure 20b) (Section 4.1). At the same time, glaciers within the 10–20° slope range still have considerable thickness and thus

volume, while higher slopes sustain thinner glaciers and much smaller volumes involved in a potential failure (Fig. 1c). Our limited data-base of detachment events suggests a transition between larger and smaller detachment volumes at roughly around

14° (Fig. 1c).

From a more mechanical point of view, low-angle glacier detachments can also be seen as part of the continuum between

surges and ice break-offs from steep glaciers. Not least owing to their slope, glaciers have a range of possibilities to adapt to changes in their stress regimes (see Fig. 22). Flat glaciers can respond by adjusting their geometry, for instance through advance

and surging. On very steep terrain, glaciers may not be able to adjust their geometry smoothly and ice breaks off. In contrast to the factors involved in typical ice break-offs from steep glaciers (see Section 1), glacier detachments appear to have in

common basal failure on soft beds. Therefore, low-angle glacier detachments combine the elements of both instability processes: the inability to rapidly adjust geometry in response to stress changes, similar to steep glaciers, and a surge-like

process that propagates an initial instability through large parts of the glacier (Thogersen et al., 2019), allowing entire glacier tongues to be mobilized. The latter framework for low-angle glacier detachments and the above one of surge endmembers are not mutually exclusive but rather linked by the role of glacier slope, fine basal tills, and the surge-like propagation of instabilities.

The role of basal water pressure in the detachments is difficult to examine in detail, but most detachments should have involved severe reduction in friction (Section 4.1), likely due to high basal water pressure. Ways to rapidly increase basal water pressure include: an increase in water input (e.g., large high-altitude rain events (Kääb et al., 2018) or increased surface snow/ice melt (Jacquemart et al., 2020)) into a subglacial drainage system not capable of adjusting fast enough; inefficiencies or blockages of this drainage system; or increased permeability of the glacier through enhanced crevassing (e.g., Tsambagarav) (Dunse et al., 2015). Sudden weakening of the strength of subglacial till under high pore-water pressure and over large parts of the glacier bed was shown to be a key process leading to the Aru detachments (Gilbert et al., 2018). Ongoing surge-like activity may enhance sensitivity to water input (Flowers et al., 2016).

In Figure 22 we attempt to summarize the main drivers of the glacier detachments described here. We define a detachment's *disposition* as the sum of long-term factors that might promote glacier detachments and refer to *triggers* to describe short-term factors that might suddenly tip the scale toward a catastrophic failure. Fundamentally, it seems that different combinations of dispositions and triggers are able to produce instability. Aside from the similarities mentioned above, the observed detachments all present some unique conditions, many of which remain shrouded in uncertainty. The observed failure conditions (Fig. 20) include ruptures during surge-like glacier instabilities (Kolka 1902, Aru, Amney Machen, Sedongpu; unclear for Devdorak) or earthquake triggered rupture (Tsambagarav), or increase of driving stress due to thickening caused by snow accumulation (Aru) or rock-ice avalanches (Kolka 2002, Amney Machen, uncertain: Sedongpu). Geothermal activity could have played a role at Tinguiririca and Kolka, but is unlikely for the other events. The thermal setting of the detached glaciers can play a role if permafrost around the glaciers potentially causes frozen margins or the glaciers exhibit a polythermal structure (Aru, Flat Creek, Tsambagarav, uncertain: Tinguirirca, Leñas). However, other detachments happened under conditions very likely free of cold ice. Some of the detached glaciers seem to have been composed of a mixture of debris and ice (Amney Machen, Flat Creek; likely at least for Devdorak, Kolka, Sedongpu). Such mixtures can be profoundly weaker than clean glacier ice, particularly at temperatures close to the melting point (Moore, 2014), but it is unclear at this point whether and how these mixtures and their weakness may have contributed to the detachments. In contrast, the ice of the Aru glaciers and the glacier at Tsambagarav clearly consisted of rather clean ice. Failure circumstances are particularly unclear to us for Aparejo and Leñas.

The angles of reach of the ice-rock avalanches associated with glacier detachments (Fahrböschung between about 5° and 10°), both absolutely and relative to their volume, are lower or at the lower end of those observed for other types of ice-rock avalanches (Fig. 1a, b). The particularly high ice content of glacier detachments might reduce friction of the mass movements through liquefaction (Schneider et al., 2011), promoting the long runouts. Remarkably, many (perhaps all) detached glaciers

appear to have sat on particularly fine-grained glacier beds. The large amounts of soft sediments under the glaciers with potentially low friction angle, combined with low effective pressure due to the presence of large amounts of basal water at the time of detachments, may have been able to reach unusually low basal shear stress. In addition, smooth u-shaped glacial valleys might favour low angles of reach by channelizing the mass flows, reducing energy dissipation, and presenting few topographic obstacles along the path (Schneider et al., 2011). The glacier detachments' long runout flows remind also of rocky "sturzstroms", some of which reached friction angles (H/L) as low as those found for the ice-rock avalanches from glacier detachments (Hsü, 1975). "Sturzstroms" and these ice-rock avalanches may share some physics, including reduced basal fraction from acoustic fluidization, or movement on air or vapour cushions of flow materials. One mechanism for long runout of dry granular flows involves acoustic fluidization (Gareth and Melosh, 2003), where the physics analogue in the long runout ice-rock avalanches may be acoustic fluidization of ice, especially in cases where the role of liquid water was mainly restricted in producing the initial mobilization of the glacier detachment.

## 4.3 Influence of climate change

Inevitably, these events rise the questionof whether climate change could be a driving factor of glacier detachments. Some cases investigated here suggest that detachments could be part of a cycle from reservoir refilling to occasional threshold exceedance, similar to what is found for glacier surges and some avalanching glaciers (Benn et al., 2019; Amney Machen, Tinguiririca, Devdorak, Kolka, uncertain: Rasht). But several cases of this study also illustrate developments where climate change can cause transient conditions that lead to failure. A number of dispositions and triggers listed in Fig. 22 may be impacted by climatic changes and may bring a glacier closer to failure, or prevent future failures, respectively.

Repeated detachments of the same glacier, or detachments connected to surge-like behaviour, require that the climate conditions and associated glacier mass balances enable reservoir recovery or built-up of accumulation areas (Devdorak, Aru, Amney Machen, Kolka 1902). Climate change might shift glaciers out of the envelope of conditions that are favourable for surging, or shift them into it (Hock et al., 2019). Whereas glacier rebuilding seems to be underway at Kolka and Aru, it is open for Sedongpu, unlikely for Tinguirica, and did not happen at all for Tsambagarav. The glacier's potential to regain a substantial size is critically linked to the potential for repeated detachment events, and thus important for hazard management.

The enhanced rock-ice avalanching onto Kolka Glacier in 2002, possibly responsible for its detachment, or the 2017 Gyala Peri rock avalanche over the Sedongpu Glacier, are likely a reflection of a general trend of climate change impact on polythermal, glacierised rock walls, where the reduction of ice cover and permafrost thaw increases the rock and ice fall frequency and enhances the potential for long-reaching cascading events (Fischer et al., 2013; Hock et al., 2019). Indeed, summers were exceptionally warm and glacier mass balances negative in the Caucasus Mountains, around Kolka Glacier, over 1998–2001 (Zemp et al., 2019). Glacier shrinkage due to negative mass balance (Larsen et al., 2015; Treichler et al., 2019; Zemp et al., 2019) has exposed large parts of the – at earlier times mostly ice-covered – headwalls of Flat Creek, Amney Machen, Leñas, and likely Petra Pervogo/Rasht. This exposes bedrock to erosion making it available for mass flows and

incorporation in and underneath the glaciers, a process that is particularly important for soft rock lithologies. The partial loss of glacier cover may also interrupt existing patterns of stress transfer and cause new temporary stress concentrations (see also Fig. 20) that may exceed stability thresholds in certain locations. Some glacier detachments could thus be connected to the transient development of headwall glacier loss.

Climate change increases the amount of meltwater and transitions from snowfall to rainfall, and may thus favour development of instabilities, at least for the polythermal glaciers (Aru and Flat Creek, uncertain: Tsambagarav) where such amounts of meltwater are unusual at the scale of the last century. The relative increase of meltwater can be particularly significant for cold/dry climate glaciers. The synchronization of the twin Aru detachments within just two months of each other points to a climatic driven instability, perhaps involving some meteorological synchronization such as exceptional amounts of high-elevation rain or snow and ice melt, or extreme weather. Also, and without understanding the triggers of the Aru events in detail, the frequency and magnitude of certain potential climatic causes and meteorological trigger events, such as heavy rain falls or warm spells, can increase with climatic changes.

Also our simplified considerations on the force balance of a glacier slab (Section 4.1) provide hints to how climate change could influence detachments. For this strongly idealized geometry, fast and extensive reduction of basal shear stress obviously reduces stability. Increase of glacier slope (e.g., from bulging), or thickness and density of the slab (e.g., additional loading from ice or rock avalanche deposition), faster than the glacier's ability to adjust, will increase driving stress. Finally, climate change could also reduce the lateral shear stress, for instance by thawing of frozen glacier margins.

## 4.4 Hazard management

From a more applied hazard management perspective, sudden massive glacier detachments pose a high magnitude/low-frequency problem, and their low probability/high consequence nature makes them hard to incorporate in hazard management and planning. The particularly low friction coefficients (H/L) involved in the detachments enable them to travel over low slopes (where other types of ice-rock avalanches would stall) and to cover large distances. The detachment events seem very rare, but their large volumes, fast evolution, and the exceptionally long runout distances and high speeds hold the potential for severe impacts even far away from the source. Our compilation of all (so far) known cases shows that low-angle glacier detachments might have, though rare, occurred more frequently than commonly thought. The differences between the events suggest that there is no straightforward way to predict where they might occur, but the following list of the most common conditions might support a more systematic assessment. Events happened at places

(i)      with abundant weak bedrocks/fine sediments,

(ii)     where glacier surface slope is between about 10° and 20°,

(iii)    where surge-like glacier instabilities are observed in the region, sometimes for the detached glacier itself
         (exception: Tsambagarav), and

(iv)    where similar events or other violent ice-rock mass flows have happened before (based on direct observation or a geomorphological imprint).

This set of very rough qualitative criteria might allow a first-order assessment of whether glacier detachments are possible in any given region. It is crucial to be aware of, however, that these criteria and their interplay are likely transient so that in particular criterion (iv) can be misleading. Particularly important for hazard management is – yet again – the conclusion that climate change is able to shift hazard zones beyond historical precedence, so that also so far unaffected areas might suddenly be susceptible (Hock et al., 2019).

For several of the events we find in aerial and satellite images streamlined debris features in the avalanche paths. Field investigations for Kolka Glacier (Petrakov et al., 2004) and Flat Creek (study in prep.) show ground traces related to the avalanche movement at two very different scales. Debris features of meters in width and tens or hundreds of meters long resemble glacial flutes, but might also consist of pavements. These features can be recognized in high-resolution aerial and satellite images and are in this study in a general way called debris stripes, acknowledging that they are not investigated in detail or understood. At a much smaller scale (mm-cm in width, dm-m in length) and thus not visible in aerial and satellite images, boulders in the avalanche path can show scratches in avalanche direction, similar to glacier striations. Notably, both scales of avalanche traces show that the ice-rock avalanches presented here were not entirely turbulent. More research is necessary to correctly interpret these signatures and differentiate them from glacial flutes, small moraines, and other longitudinal glacial and geomorphodynamic features.

Systematic monitoring turns out to be one of the most feasible responses to changes in hazard conditions, and the increasing temporal and spatial resolution, and improving availability of satellite imagery, is particularly helpful for glacier detachment assessment. For several of the cases in this contribution, abnormal glacier crevassing and accelerating precursory speeds were visible days to weeks before failure (Kolka 2002, Rasht, Aru, Sedongpu, Tsambagarav, uncertain: Flat Creek), but the significance of such a development was realized for the second Aru detachment only due to its spatial and temporal proximity to the first Aru detachment. On the other hand, abnormal crevassing does not necessarily indicate an impending detachment, see Rasht (situation in 2007, Supplementary Fig. S2) or Leinss et al. (2019), where the glacier geometry has likely stabilized a detaching ice mass. Overall, predicting glacier detachments can likely only be achieved by strong efforts in detailed remote-sensing based monitoring and, if feasible, by ground-based measurements, which contribute to an improved understanding of conditions and relevant processes, past events, glacier velocities and slope deformations, glacier bed geology and lithology, surge behaviour and dynamics, and short-term and long-term temperature and precipitation records (Kääb et al. 2018).

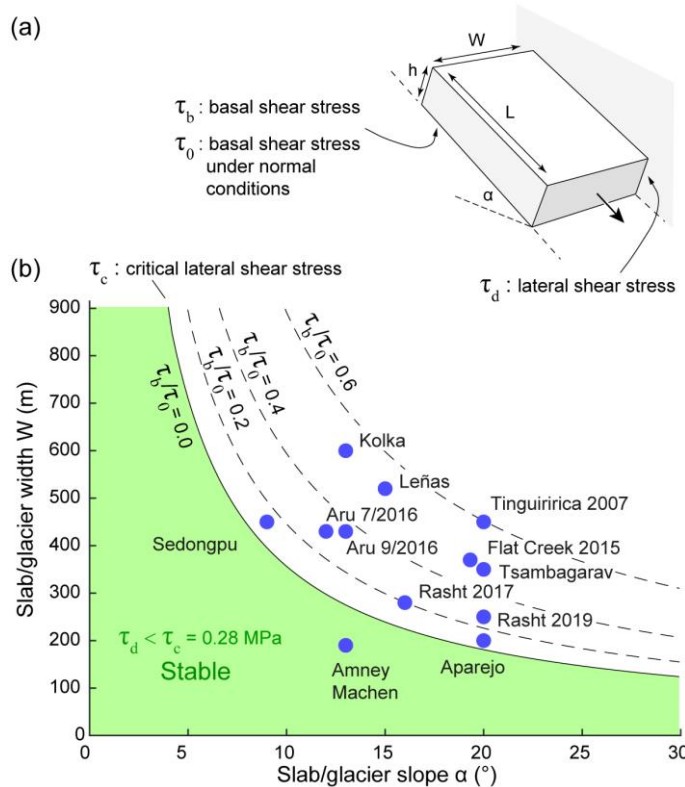

Figure 20: (a) Slab geometry used for the force balance analysis. (b) Stability diagram showing under which condition the slab
would remain stable in the case of a total loss of basal friction( $\tau_b = 0$ ; green area) and for different ratios friction $\tau_b/\tau_0$ (i.e.
partial loss of basal friction; dashed lines). Blue dots show averaged slope and width of all detachment reported in this study.
The critical lateral shear stress $\tau_c$ has been estimated from the Aru glacier detachments using the force balance constructed in
Gilbert et al. (2018).

| Name of event | Earlier surges in the region? | Weak rocks, fine sediments ? | Glacier tongue within permafrost ? | Abnormal crevassing before failure? | Enhanced water input into glacier (melt, rain) ? | Repeat events known? | Signs of high water pressure ? | Signs of earlier mass flows? | Surge-like activity related to event? | Loading prior to event (snow, rock)? | Earlier surges same glacier ? | Bulging before event? |
|---|---|---|---|---|---|---|---|---|---|---|---|---|
| Kolka | | | | | | | | | | | | |
| Devdorak | | | | | | | | | | | | |
| Aru | | | | | | | | | | | | |
| Sedongpu | | | | | | | | | | | | |
| Rasht | | | | | | | | | | | | |
| Amney Machen | | | | | | | | | | | | |
| Flat Creek | | | | | | | | | | | | |
| Tinguiririca | | | | | | | | | | | | |
| Aparejo | | | | | | | | | | | | |
| Leñas | | | | | | | | | | | | |
| Tsambagarav | | | | | | | | | x | | | |

| yes | yes, but uncertain | unknown | no, but uncertain | no | x Earthquake-triggered rupture ca. 2 weeks before detachment |

 **Figure 21: Possible indicators for and factors involved in low-angle glacier detachments.**

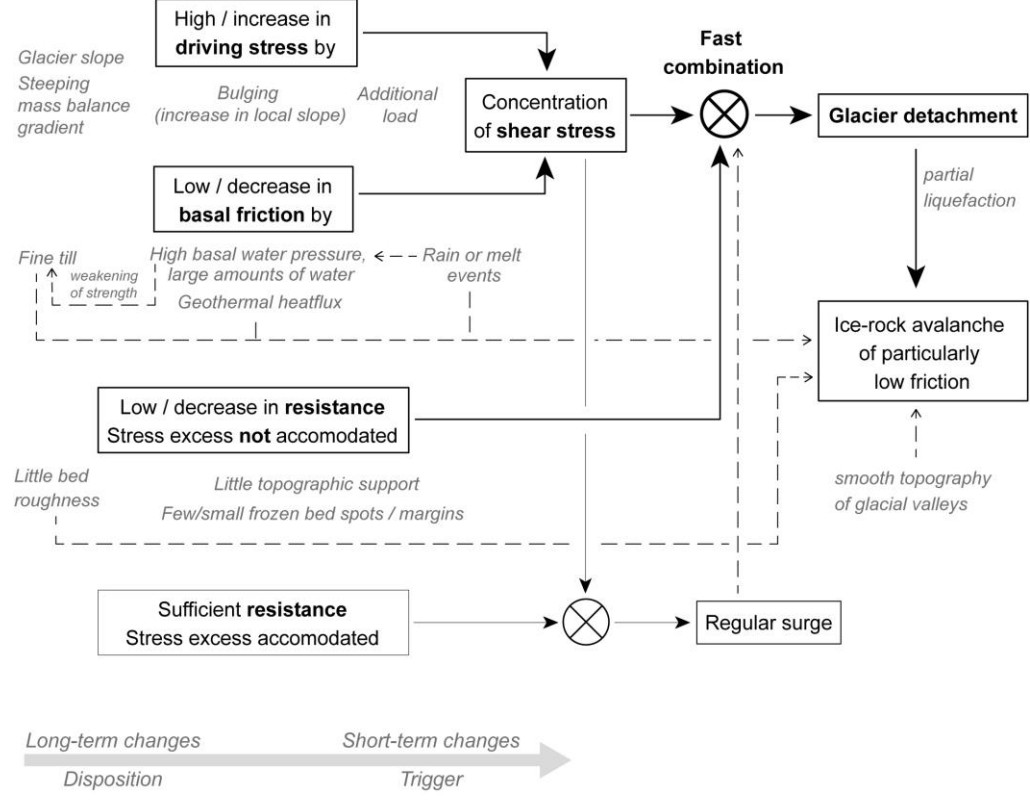

 **Figure 22: Schematic on which conditions and changes to a low-angle mountain glacier can in combination lead to the spatio-temporal interference of particularly high and concentrated shear stresses and low resistance, eventually exceeding stability thresholds and causing detachment. The failure conditions can change at a range of time scales, so that detachment is the result of a highly transient and rare interplay of factors. It is key that the low basal friction and high driving stresses, the resulting concentration of high shear stresses, and the lack of sufficient resistance develop rapidly, or are combined rapidly, preventing the glacier to adjust to changing forces in a steady way. Several of the factors potentially involved in the detachment are subsequently also able to strongly reduce basal friction of the resulting ice-rock avalanche and lead thus to particularly low angles of reach. Boxes in the figure indicate main physical conditions, grey italic text indicates different actual processes that can fulfil these conditions, sorted from long-term (left) to short-term (right) variability.**

# 5 Conclusions

In this contribution we describe around 20 ice-rock avalanche events that we characterize as sudden large-volume detachments of low-angle glaciers. Overall, these events seem to be more frequent than previously thought. The detached volumes ranged

from a few up to more than $100 \cdot 10^6$ m$^3$. We described one new event in the same size-class as the 2002 Kolka and the 2016 Aru glacier detachments (Sedongpu 2018) and as a side-result quantified one of the larger high-mountain rock avalanches of recent decades (Gyala Peri 2017).

Despite the relatively low number of low-angle glacier detachments and their site-specific variations that leave considerable uncertainties, we were able to identify a set of conditions and evolutions likely involved in these glacier failures. We consider this an important step, given that a few years ago the possibility for low-angle mountain glaciers to detach and produce massive ice-rock avalanches was hardly known. Interestingly, the fact that the spatio-temporal factor combinations leading to exceptionally low basal friction and very high shear stress concentration, and eventually to detachment, are different among the cases, suggests that there exists an exceptional but still fundamental possibility of low-angle glacier beds to fail catastrophically. Awareness of this fundamental potential for catastrophic basal instability expands our understanding of glacier flow.

Many of the glacier detachments show some relation to surge-type glacier movement, and could be seen as a rare and extreme endmember of this much more common glacier instability. Glacier detachments combine elements of surging, where the glacier adjusts its geometry to satisfy the force balance, with those of ice break-offs from steeper glaciers, where the glacier is not able to adjust in a steady way. The surface slopes of 9–21° of the detached glacier parts, though quite low for glaciers that produce ice avalanches, might rank high in comparison with surge-type glacier tongues. Slopes in this range exert higher shear stresses than is typical for surge-type glaciers and favour thus the possibility of exceeding a critical stress level that then leads to sudden failure. At the same time, in comparison to glaciers in very steep terrain that tend to be thin, glaciers of 10–20° surface slope can build up thicker ice, which leads to the larger volumes typically involved in detachments. Using a strongly simplifying slab model we estimate ranges for glacier slope and width above which a glacier could detach when widespread and strong loss of its basal resistance cannot be accommodated anymore by lateral resistance. We estimate a critical shear stress of 0.28 MPa that could be supported by the glacier margins of the Aru detachments. We also estimate that (surge-like) precursory acceleration of glacier sliding before detachment could produce substantial amounts of ice melt at the glacier base that would facilitate reduction of the shear stress of soft beds.

Weak bedrock and/or the existence of soft and highly erodible sediments under the detached glaciers was identified for most of the observed glacier detachments, plausible for the ones we determined retroactively, and hints at till-strength weakening under high pore-water pressure as a concrete failure process. There appear to be different trajectories into, or out of, the narrow envelope of potential failure conditions, not least driven by climatic changes. The fact that most collapses happened during local spring/summer suggests that a meltwater or high-altitude rain driven increase of basal pore water pressure can play an important role in triggering the events, whether directly or in some delayed form. Atmospheric warming enhances such hazard conditions. For some detachment sites, negative regional glacier mass balances can prevent detached glaciers from fully rebuilding and thus from detachments to repeat over time, at least at earlier volumes. The special case of Tsambagarav

demonstrates that an earthquake was not able to trigger a glacier detachment (reinforcing earlier findings that low-slope glaciers appear quite resistant to ground shaking; Kargel et al., 2016) but to precondition one glacier for failure along a weakness existing already before the earthquake.

Detailed investigations of the events described in this study showed that a wide variety of dispositions and triggers can lead to a glacier detachment. This makes it challenging for practitioners working in high-mountain hazard management to anticipate and predict such events. From this practical standpoint, however, this study attempts to raise awareness about the – albeit low – possibility of sudden, large-volume detachments of low-angle glaciers at locations with

- particularly soft and erodible lithologies, and, likely related, the
- existence of surge-type glaciers and surge-like glacier evolution.
- Repeated events or geomorphological imprints of potential earlier collapses or other violent ice-rock mass flows can be further investigated, but events can also happen without historical precedence through shifts in the array of failure conditions.
- Several of the glaciers investigated here showed abnormal crevassing and enhanced precursory surface speeds in the days to weeks before detachment.
- The surface slopes found in this study for the detached glaciers ranged between roughly 10° and 20°, and we propose a rough combination of glacier slope and width above which glaciers could detach in case of extensive loss of basal friction (section 4.1).

Due to the large amounts of snow and ice involved in glacier detachments, the high chance of lubrication and of liquefaction of glacier ice and subglacial sediments, and smooth geometries of glacial valleys, avalanche friction is typically greatly reduced. This results in the particularly high mobility of the ice-rock avalanches resulting from low-angle glacier detachments and can lead to substantial damage far from the source. Between the large runout distances and the varying factors that can impact a glacier's detachment probability, high-mountain hazard management will, after the first general assessment provided in this study, benefit from more detailed investigations of glacier detachments, the conditions that lead to them and the mechanics that drive them.

**Table 1: Parametres for the low-angle glacier detachments of this contribution. For repeat events the parameters given refer to the underlined year.**

| Name of event | Region | Years | Lat, Lon | GLIMS,RGI IDs | Reach L (km) | Drop H (km) | Volume (10⁶ m³) | Glacier surface slope (°) | Section |
|---|---|---|---|---|---|---|---|---|---|
| Devdorak | Caucasus | (1776, 1778, 1785, 1808, 1817 ?)1832, 2014 | 42.72°N, 44.55°E | G044517E42715N, RGI60-12.00840 | 10.5 | 3.2 | 15 | 17(?) | 3.1 |
| Kolka | Caucasus | ~1835?,1902, 2002 | 42.73°N, 44.44°E | G044447E42733N, RGI60-12.00131 | 19 | 2.0 | 130 | 13 | 3.1 |
| Rasht | Pamirs | 2017 | 38.975N 70.850E | G070852E38974N, RGI60-13.18284; USSR 504 | 7.6 | 1.4 | 6 | 16 | 3.2 |
| Rasht | Pamirs | 2019 | 38.989° N 70.693° E | G070689E38981N, RGI60-13.20645; USSR 518 | 6.7 | 1,5 | 4.5 | 20 | 3.2 |
| Aru 1 | Western Tibet | Jul 2016 | 34.02° N, 82.250° E | G082249E34023N, RGI RGI60-13.51473, Chinese: CN5Z412C0011 | 8.2 | 0.8 | 68 | 12 | 3.3 |
| Aru 2 | Western Tibet | Sep 2016 | 34.00° N, 82.265° E | G082268E34005N, RGI60-13.51476, Chinese inv.: CN5Z412C0007 | 7.2 | 0.83 | 83 | 13 | 3.3 |
| Tsambagarav | Western Mongolia, Altai | Aug 1988 | 48.66° N, 90.75° E, | G090733E48676N, RGI60-10.02686 | 4-5.5 | 0.7 | 6 | 20 | 3.4 |
| Amney Machen | Eastern Tibet | 2004,2007, 2016,2017,2019 | 34.82°N 99.44°E | G099443E34824N, RGI60-13.23943, Chinese inv: CN5J352E0017, | 5.2 | 1.0 | 27 | 13 | 3.5 |
| Sedongpu | South-eastern Tibet | 2018 | 29.80° N 94.92° E | G094940E29811N, RGI60-13.01391, Chinese inv.: SDP1 | 8 | 1.3 | 100 | 9 | 3.6 |
| Flat Creek | Saint Elias Mts. | 2013, 2015, 2016 | 61.50° N, 141.54° W | G218441E61638N, RGI60-01.17460 | 11 | 1.1 | 14 | 20 | 3.7 |
| Aparejo | Chilean Andes | 1980 | 33.56° S, 70.01° W | only World Glacier Inv. CL1M005D0049 | 3.7 | 0.8 | 7.2 | 20 | 3.8 |
| Leñas | Argentinean Andes | 2007 | 34.46° S 70.05° W | G289941E34459S, RGI60-17.01251 | 2 | 0.2 | 4.2 | 15 | 3.9 |

| Tinguiririca | Chilean Andes | ~1960,1994, 2007 | 34.83° S 70.35° W | only World Glacier Inv. CL1M00420016 | 7.9 | 1.4 | 12 | 20 | 3.10 |

## Code availability

 Not relevant

## Data availability

 Sentinel-1 and Sentinel-2 data are freely available from the ESA/EC Copernicus Sentinels Scientific Data Hub at https://scihub.copernicus.eu (Copernicus, 2020), Landsat satellite data, Corona satellite data, and SRTM-C DEMs from United  States Geological Survey (USGS, 2020), SRTM-X and TanDEM-X DEMs from the German Aerospace Center (DLR, 2020), the ALOS World DEM from the Japan Aerospace Exploration Agency (JAXA, 2020), Chinese earthquake data from the China  Earthquake Data Center. Planet data (Dove and RapidEye) are not openly available as Planet is a commercial company. However, scientific access schemes to these data exist (https://www.planet.com/markets/education-and-research/). Data from  Maxar satellites (GeoEye, Ikonos, WorldView, Quickbird) and Airbus (Pléiades, Spot) are commercial, but in parts explorable from GoogleEarth and BingMaps.

 ## Author contribution

A.K. wrote the text, did most new analyses and prepared the figures. M.J. and A.G. wrote parts of the text, edited all text, and  contributed comments and discussions. L.G. prepared the Sedongpu DEMs. All authors edited text and contributed with data, comments and discussions.

 ## Competing interests

All authors declare that they have no competing interests.

 ## Acknowledgements

This paper is an extended and updated version of the Louis Agassiz medal lecture by A.K. given at the European Geosciences 1102 Union General Assembly 2019. We would like to thank Martin Truffer and an anonymous referee for their very constructive comments. The force balance model, section 4.1, is based on a concept proposed to us by Martin Truffer. We are grateful to  the providers of free data for this study; European Space Agency (ESA) / European Commission (EC) Copernicus for Sentinel-

2 data, USGS for Landsat, Corona, and SRTM data, DLR for SRTM and TanDEM-X data, and Planet for their cubesat data via Planet's Ambassadors Program. E.B. and S.G. acknowledge support from the French Space Agency (CNES) through TOSCA and DINAMIS programs. S.C. acknowledges support from Lomonosov Moscow State University on the theme "Mapping, modeling and risk assessment of dangerous natural processes". D.P. and S.C. acknowledge the Russian Foundation for Basic Research. Paola Banegas from SEGEMAR Mendoza provided the 1962 aerial photos over Tinguririca.

## Financial support

This work was funded by the ESA projects Permafrost_CCI (4000123681/18/I-NB), Glaciers_CCI (4000109873/14/I-NB, 4000127593/19/I-NS), and the ESA EarthExplorer10 Mission Advisory Group (4000127656/19/NL/FF/gp), by the European Research Council under the European Union's Seventh Framework Programme (FP/2007-2013) / ERC grant agreement no. 320816, and the Russian Foundation for Basic Research (grant 18-05-00520). JSK thanks NASA's Interdisciplinary Science Program (grant 80NSSC18K0432).

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
