# Peer review of "Sudden large-volume detachments of low-angle mountain glaciers — more frequent than thought?"

_The Cryosphere, 2020_

## Referee Comment (RC1) · Anonymous Referee #1 · 16 Nov 2020

General comments:

The manuscript reviews exiting and presents new information on glacier instabilities with a focus on glacier detachments. Observations on glacier detachments are extremely rare and, hence, our knowledge is very limited. The manuscript presents a model for glacier detachments and ideas for a first assessment of related hazards. This well-written text is of high importance and as such a welcome contribution.

Specific comments:

(1) The title of the manuscript is "Sudden large-volume detachments of low-angle mountain glaciers – more frequent than thought" and, hence, stresses the higher-than

expected frequency as THE main conclusion of the study; otherwise, the second part of the title could be deleted. Throughout the manuscript, the authors repeatedly mention that glacier detachment events are rare and occur more often than thought:

L 80-81: "Our third type of glacier instability, a sudden large-scale detachment of a mountain glacier, is much less frequent than the two types above, [...]" L 804-805: "Our compilation of all (so far) known cases shows that low-angle glacier detachments might have, though rare, more frequently occurred than thought." L845: "Overall, these events seem to be more frequent than previously thought."

However, there is no explanation of why these events should occur (significantly) more often than has been reported. If the explanation is simply "because we will find additional ones once we look for them," the second part of the title is misleading as it suggests the presentation of facts rather than a simple assumption. Yes, of course, it should be assumed that there are more of them because we recently learned about them. I like to suggest the deletion of the second part of the title–ĂŤit seems to aim at a dramatization that is unnecessary here.

(2) The authors aim to introduce glacier detachment as a new type of glacier instability that is different from glacier surges and from ice-rock avalanches. While the authors introduce an entire chapter on ice-rock avalanches with the goal of clearly separate them from glacier detachments, they do not have a similar chapter on glacier surges. It is not clear, why the ice-rock avalanches receive much more attention. As a solution, Chapter 2 Ice-rock avalanches could be deleted, or the authors shorten it by deleting the list of examples (references are sufficient) and then add a similar chapter on glacier surges.

(3) It is not clear what the prove is for classifying the 1832 Devdorak and 1902 Kolka events as glacier detachments. Please revise.

(4) L 234 introduces the term "striations" for streamlined debris stripes. This is unfortunate as the term glacial striations is already defined. Hence, using the term striation in

a paper on glaciers is confusing. The term "debris stripes" is sufficient. Please change this throughout the manuscript.

(5) L 775-776: The possibility of a meteorological trigger that resulted in the failure of the Aru glaciers does not necessarily point at climate change. Please be careful with such assumptions. We are not even certain that there was a meteorological event at all.

Technical corrections:

L 170: delete "Devdoraki in Georgian language" as you don't present local names for all other cases. L 349: you mention "Before-after elevation differences are available in Kääb et al. (2018)." Either simply present them here in one sentence or delete the sentence. L 825: write Rasht (2007). Enlarge the following figures (take advantage of entire width; see, e.g., Fig. 5): 2, 3, 4, 6, 7, 8, 13, 15, 16, 17, 18.

---

## Referee Comment (RC2) · Martin Truffer (Referee) · 20 Dec 2020

This manuscript addresses a glacier phenomenon that was until recently almost entirely unknown in the glaciological community. The authors make the case that the catastrophic disintegration of large parts of relatively low-sloped glaciers is perhaps not as uncommon as previously thought. The paper presents a collection of events, some of which are reported here for the first time. This paper is very important in that it provides a first base line for something that is glaciologically interesting, but that has very important implications in terms of natural hazard planning. While the glacier detachment events reported here are rare, they can be devastating, as shown by the Aru

events, and perhaps they also occur with sufficient warning to prevent the loss of human life, as shown by the second Aru event. The authors make a first assessment as to what conditions could lead to such detachments, and place the phenomenon as something between the better known occurrences of glacier surging and ice avalanching of hanging glaciers. The manuscript is also a testament to the power of modern remote sensing tools in reconstructing these events, in terms of just imagery, but also with repeat DEMs and feature tracking for volume and velocity evolution. The manuscript is well written and it can essentially be published as is. I am attaching a marked up copy with the very few editorial comments I have.

This manuscript is also thought provoking, and I would personally welcome some additions to the Discussion. This is more a matter of taste, so I would leave it to the authors/editors whether such an expansion is warranted. But in a second attachment I tried to line out some simple force balance considerations that could potentially be useful in helping explain the phenomenon. In most ways, they just expand on explanations already in the manuscript. In particular, a very simplified view of a glacier as a slab of rectangular cross section provides some insight: There is a maximum stress that can be imposed on the sides (which is reached when basal stress is zero), and if that maximum stress exceeds the failure strength of ice, then catastrophic failure must occur. This stress depends on the slope of the glacier and also on the half-width to depth ratio. In particular, it explains why low sloped glaciers cannot detach. Furthermore, this implies that relatively straight glaciers are more prone to large scale failures, because curvature along flow provides an additional mechanism to accommodate stress. Also, a quick calculation of the effect of rock falls on glaciers show that in most cases, the increase in driving stress is larger than the increase in effective pressure, which would imply that rock falls move a glacier closer to an instability. This is especially true, if basal water pressures can evolve into a direction to decrease effective pressure (which sees an initial increase due to the additional overburden). Finally, one thing that could be discussed in terms of the observation of fine grained sediments: Fine grained beds are (in my opinion) more likely to allow conditions where essentially zero effective stress can be attained over large areas, particularly with distributed sources of water supply (such as excess geothermal heat). On a solid bedrock, widespread high water pressure will lead to extensive bed separation, which eventually leads to channelized water flow and rapid de-watering of the bed. This can be more delayed on sediment beds, where much high-pressure water can be retained in the sediment structure.

It might be beyond the scope of this paper to add such considerations, but I find them interesting, which is why I added them here.

Martin Truffer

Please also note the supplement to this comment:
https://tc.copernicus.org/preprints/tc-2020-243/tc-2020-243-RC2-supplement.pdf

**Fig. 1.**

**Supplement:**

[revised manuscript text omitted]

**Simple force balance**

$$HWggg\sin\alpha = W\tau_b + 2H\tau_s$$

$$\tau_s = \left(\frac{W}{2H}\right)ggg\sin\alpha - \frac{W}{2H}\tau_b$$

half-width to depth ratio slope

Maximum side stress occurs for $\tau_b = 0$

If $\tau_{s,max} \geqslant$ failure strength

$\Rightarrow$ detachment otherwise : rapid flow or surge till bed : $\tau_b = N \cdot \tan\phi$

$\tan\phi$ smaller for fine grained till hard bed : more unlikely to achieve $N \doteq 0$ on large spatial scale because increased bed separation would lead to more pathways for water to drain soft bed : large scale areas of low $N$ are easier to achieve, particularly combined with distributed water sources, such as increased geothermal heat

**Influence of sudden mass addition (rock fall)**

$$\Delta\tau_d = g_r h_r\, g\sin\alpha$$

$$\Delta\tau_b = \underbrace{g_r h_r\, g\cos\alpha}_{\Delta N} \cdot \tan\phi$$

$$\left. \right\} \quad \frac{\Delta\tau_b}{\Delta\tau_d} = \frac{\tan\alpha}{\tan\phi}$$

If $\Delta\tau_b/\Delta\tau_d < 1$ then rock fall leads to more unstable configuration

Also, over time, if water pressure increases and $N$ decreases, instability can occur

---

## Author Response (AR1)

tc-2020-243

**Sudden large-volume detachments of low-angle mountain glaciers – more frequent than thought**

Andreas Kääb, Mylène Jacquemart, Adrien Gilbert, Silvan Leinss, Luc Girod, Christian Huggel, Daniel Falaschi, Felipe Ugalde, Dmitry Petrakov, Sergey Chernomorets, Mikhail Dokukin, Frank Paul, Simon Gascoin, Etienne Berthier, Jeff Kargel

**Revisions in response to referees**

**General response**

We would like to thank the two referees for their positive and very constructive reviews that certainly helped to improve the paper! We modified our manuscript as described below in detail. To summarize, as the largest changes, we modified section 2 to contain also relevant background on surges, we add a short section with a case not mentioned yet in the initial manuscript, and we add a subsection with a simplified force-balance estimation for detachments and related energy balance considerations. These changes are also highlighted in the attached track-changes document.

**As we were unsure whether the manuscript will go out again to referee 2 (recommendation "technical corrections") we took the freedom to send the revised manuscript to him for check of the new parts he suggested (section 4.1). He had no objections.**

We also went carefully again through the entire manuscript and corrected typos and edited language and grammar details. These small changes are not contained in the mark-up for clarity but we would be happy to provide such a complete change document upon request.

Referee comments are in *italic*, and our response and what revisions we will do in normal font.

**Response to individual referees**

Referee 1: page 1

Referee 2: page 3

**Referee #1, anonymous**

*General comments:*

*The manuscript reviews exiting and presents new information on glacier instabilities with a focus on glacier detachments. Observations on glacier detachments are extremely rare and, hence, our knowledge is very limited.*

*The manuscript presents a model for glacier detachments and ideas for a first assessment of related hazards. This well-written text is of high importance and as such a welcome contribution.*

*Specific comments:*

*(1) The title of the manuscript is "Sudden large-volume detachments of low-angle mountain glaciers – more frequent than thought" and, hence, stresses the higher-than expected frequency as THE main conclusion of the study; otherwise, the second part of the title could be deleted. Throughout the manuscript, the authors repeatedly mention that glacier detachment events are rare and occur more often than thought:*

*L 80-81: "Our third type of glacier instability, a sudden large-scale detachment of a mountain glacier, is much less frequent than the two types above, [...]" L 804-805: "Our compilation of all (so far) known cases shows that low-angle glacier detachment smight have, though rare, more frequently occurred than thought." L845: "Overall, these events seem to be more frequent than previously thought."*

*However, there is no explanation of why these events should occur (significantly) more often than has been reported. If the explanation is simply "because we will find additional ones once we look for them," the second part of the title is misleading as it suggests the presentation of facts rather than a simple assumption. Yes, of course, it should be assumed that there are more of them because we recently learned about them. I like to suggest the deletion of the second part of the title seems to aim at a dramatization that is unnecessary here.*

We had much discussion about the title among the author team and there was a clear vote to keep the second part of the title, supported also by referee 2. We now put a question mark behind this part of the title to show that we don't mean a fact.

*(2) The authors aim to introduce glacier detachment as a new type of glacier instability that is different from glacier surges and from ice-rock avalanches. While the authors introduce an entire chapter on ice-rock avalanches with the goal of clearly separate them from glacier detachments, they do not have a similar chapter on glacier surges. It is not clear, why the ice-rock avalanches receive much more attention. As a solution, Chapter 2 Ice-rock avalanches could be deleted, or the authors shorten it by deleting the list of examples (references are sufficient) and then add a similar chapter on glacier surges.*

We shorten the ice-rock avalanche overview a bit and add context regarding glacier surging, in particular less common aspects of surging that appear, however, especially interesting in the context of detachments.

*(3) It is not clear what the prove is for classifying the 1832 Devdorak and 1902 Kolka events as glacier detachments. Please revise.*

We clarify that the 1902 Kolka event certainly was a detachment, but that such conclusion for the 1832 Devdorak event is well possible but uncertain.

*(4) L 234 introduces the term "striations" for streamlined debris stripes. This is unfortunate as the term glacial striations is already defined. Hence, using the term striation in a paper on glaciers is confusing. The term "debris stripes" is sufficient. Please change this throughout the manuscript.*

Agreed and changed; we are actually also more comfortable with "debris stripes". In section 4.4 we add a small paragraph about these features.

*(5) L 775-776: The possibility of a meteorological trigger that resulted in the failure of the Aru glaciers does not necessarily point at climate change. Please be careful with such assumptions. We are not even certain that there was a meteorological event at all.*

Agreed. We formulate now more careful as a possibility, and move the thought to a less prominent location at the end of the sub-section.

*Technical corrections:*

*L 170: delete "Devdoraki in Georgian language" as you don't present local names for all other cases.* We kept as about half of the literature, web entries, news, etc, on this case use the Georgian name, and we want facilitate internet search by readers.

*L 349: you mention "Before-after elevation differences are available in Kääb et al. (2018)." Either simply present them here in one sentence or delete the sentence.* Specified

*L 825: write Rasht (2007).* Clarified

*Enlarge the following figures (take advantage of entire width; see, e.g., Fig. 5): 2, 3, 4, 6, 7, 8, 13, 15, 16, 17, 18.* We will make sure in the production phase.
* * *
**Referee #2 Martin Truffer**

*This manuscript addresses a glacier phenomenon that was until recently almost en-tirely unknown in the glaciological community. The authors make the case that the catastrophic disintegration of large parts of relatively low-sloped glaciers is perhaps not as uncommon as previously thought. The paper presents a collection of events, some of which are reported here for the first time. This paper is very important in that it provides a first base line for something that is glaciologically interesting, but that has very important implications in terms of natural hazard planning. While the glacier de-tachment events reported here are rare, they can be devastating, as shown by the Aru events, and perhaps they also occur with sufficient warning to prevent the loss of hu-man life, as shown by the second Aru event. The authors make a first assessment as to what conditions could lead to such detachments, and place the phenomenon as some-thing between the better known occurrences of glacier surging and ice avalanching of hanging glaciers. The manuscript is also a testament to the power of modern remote sensing tools in reconstructing these events, in terms of just imagery, but also with repeat DEMs and feature tracking for volume and velocity evolution. The manuscript is well written and it can essentially be published as is. I am attaching a marked up copy with the very few editorial comments I have.*

All editorial comments were implemented.

*This manuscript is also thought provoking, and I would personally welcome some ad-ditions to the Discussion. This is more a matter of taste, so I would leave it to the authors/editors whether such an expansion is warranted. But in a second attachment I tried to line out some simple force balance considerations that could potentially be use-ful in helping explain the phenomenon. In most ways, they just expand on explanations already in the manuscript. In particular, a very simplified view of a glacier as a slab of rectangular cross section provides some insight: There is a maximum stress that can be imposed on the sides (which is reached when basal stress is zero), and if that maximum stress exceeds the failure strength of ice, then catastrophic failure must oc-cur. This stress depends on the slope of the glacier and also on the half-width to depth ratio. In particular, it explains why low sloped glaciers cannot detach. Furthermore, this implies that relatively straight glaciers are more prone to large scale failures, be-cause curvature along flow provides an additional mechanism to accommodate stress. Also, a quick calculation of the effect of rock falls on glaciers show that in most cases, the increase in driving stress is larger than the increase in effective pressure, which would imply that rock falls move a glacier closer to an instability. This is especially true, if basal water pressures can evolve into a direction to decrease effective pressure(which sees an initial increase due to the additional overburden). Finally, one thing that could be discussed in terms of the observation of fine grained sediments: Fine grained beds are (in my opinion) more likely to allow conditions where essentially zero effective stress can be attained over large areas, particularly with distributed sources of water supply (such as excess geothermal heat). On a solid bedrock, widespread high water pressure will lead to extensive bed separation,*

*which eventually leads to channelized water flow and rapid de-watering of the bed. This can be more delayed on sediment beds, where much high-pressure water can be retained in the sediment structure.*

*It might be beyond the scope of this paper to add such considerations, but I find them interesting, which is why I added them here.*

We would like to thank the referee a lot for his detailed reflections, and his concept for a force balance. We added a new subsection and figure in the discussion about a simplified force and energy balance, and adapt the text else at some places in line with above thoughts.